

# Criteria for the optimal selection of remote sensing optical images to map event landslides

Fiorucci F.[1], Giordan D.[2], Santangelo M.[1], Dutto F.[3], Rossi M.[1], Guzzetti F.[1]

1   Istituto di Ricerca per la Protezione Idrogeologica, Consiglio Nazionale delle Ricerche, via della Madonna Alta 126, 06128 Perugia, Italy

2   Istituto di Ricerca per la Protezione Idrogeologica, Consiglio Nazionale delle Ricerche, Strada delle Cacce 73, 10135 Torino, Italy

3   Servizio Protezione Civile della Città Metropolitana di Torino, Via Alberto Sordi 13, 10095 Grugliasco, Italy

Correspondence to: Federica Fiorucci (Federica.Fiorucci@irpi.cnr.it)

**Abstract**
We executed an experiment to determine the effects of optical image characteristics on event
landslide mapping. In the experiment, we compared eight maps of the same landslide, the
Assignano landslide, in Umbria, Central Italy. Six maps were obtained through the expert visual
interpretation of monoscopic and pseudo-stereoscopic (2.5D), ultra-resolution ($3 \times 3$ cm) images
taken on 14 April 2014 by a Canon EOS M photographic camera flown by an CarbonCore 950
hexacopter over the landslide, and of monoscopic and stereoscopic, true-colour and false-colour-
composite, $1.84 \times 1.84$ m resolution images taken by the WorldView-2 satellite also on 14 April
2014. The seventh map was prepared through a reconnaissance field survey aided by a pre-event
satellite image taken on 8 July 2013, available on Google Earth™, and by colour photographs taken
in the field with a hand-held camera. The images were interpreted visually by an expert
geomorphologist using the StereoMirror™ hardware technology combined with the ERDAS
IMAGINE® and Leica Photogrammetry Suite (LPS) software. The eighth map, which we
considered our reference showing the "ground truth", was obtained through a Real Time Kinematic
Differential Global Positioning System (GPS) survey conducted by walking a GPS receiver along
the landslide perimeter to capture geographic coordinates every about 5 m, with centimetre
accuracy. The eight maps of the Assignano landslide were stored in a Geographic Information
System (GIS), and compared adopting a pairwise approach. Results of the comparisons, quantified
by the error index $E$, revealed that where the landslide signature was primarily photographical (in
the landslide source and transport area) the best mapping results were obtained using the higher
resolution images, and where the landslide signature was mainly morphometric (in the landslide
deposit) the best results were obtained using the stereoscopic images. The ultra-resolution image
proved very effective to map the landslide, with results comparable to those obtained using the
stereoscopic satellite image. Conversely, the field-based reconnaissance mapping provided the
poorest results, measured by large mapping errors, and confirmed the difficulty in preparing
accurate landslide maps in the field. Albeit conducted on a single landslide, we maintain that our
results are general, and provide useful information to decide on the optimal imagery for the
production of event, seasonal and multi-temporal landslide inventory maps.

## 1 Introduction

Accurate detection of single landslides has different scopes, including landslide mapping (Di Maio and Vassallo, 2011; Manconi et al., 2014; Plank et al., 2016), landslide hazard analysis and risk assessment (Allasia et al., 2013), to support the installation of landslide monitoring systems (Tarchi et al., 2003; Teza et al., 2007; Monserrat and Crosetto, 2008; Giordan et al., 2013), and for landslide geotechnical characterization and modelling (Gokceoglu, 2005; Rosi et al., 2013). Mapping of single landslides can be executed using the same techniques and tools commonly used by geomorphologists to prepare landslide inventory maps i.e., through field surveys (Santangelo et al., 2010) or the heuristic visual interpretation of monoscopic or stereoscopic aerial or satellite images (Brardinoni et al., 2003; Fiorucci et al., 2011; Ardizzone et al., 2013), of LiDAR-derived images (Ardizzone et al., 2007; Van Den Eeckhaut et al., 2007; Haneberg et al., 2009; Giordan et al., 2013; Razak et al., 2013; Niculita et al., 2016, Petschko et al., 2016 ), or of ultra-resolution images acquired by Unmanned Aerial Vehicles (UAV, Niethammer et al., 2010, Giordan et al., 2015a, 2015b; Torrero et al., 2015, Turner et al., 2015). The heuristic visual mapping of landslide features is based on the systematic analysis of image photographic and morphological characteristics such as colour, tone, mottling, texture, shape, size, curvature (Pike, 1988). These photographic and morphological characteristics encompasses all the possible landslide features that can be used for the (visual) interpretation of the available imagery.

All these mapping techniques have inherent advantages and intrinsic limitations, which depend on the size and type of the landslides, and on the characteristics of the images, including their spatial and spectral resolutions (Fiorucci et al., 2011). As a result, landslide maps prepared exploiting one or more of the mentioned techniques are inevitably incomplete, and contain errors in terms of the position, size and shape of the mapped landslides (Guzzetti et al., 2000; Galli et al., 2008, Santangelo et al., 2015a).

Attempts have been made to evaluate the errors associated to different types of landslide inventory maps (Carrara et al., 1992; Ardizzone et al., 2002, 2007; Van Den Eeckhaut et al., 2007; Fiorucci et al., 2011; Santangelo et al., 2010; Mondini et al., 2013). Most of these attempts compare landslide maps prepared using aerial or satellite images to maps obtained through reconnaissance field mapping (Ardizzone et al., 2007; Fiorucci et al., 2011) or GPS surveys (Santangelo et al., 2010). Conversely, only a few authors have attempted to evaluate the influence of different types

of imagery on landslide detection and mapping (Carrara et al., 1992).
In this work, we evaluate how images of different types and characteristics influence event
landslide mapping. We do this by comparing eight maps of a single, rainfall-induced landslide near
the village of Assignano, Umbria, central Italy. Seven maps of the same landslide were obtained
using different techniques and images, including (i) a reconnaissance field survey, (ii) the
interpretation of ultra-resolution images taken by an optical camera on-board an UAV, and (iii) the
visual interpretation of Very High Resolution (VHR), monoscopic and stereoscopic, multispectral
images taken by the WordView-2 satellite. These maps were compared to an eighth map considered
to be the benchmark showing the "ground truth" i.e., the "true" position, shape and extent of the
Assignano landslide. Based on the results of the map comparison, we infer the ability of different
optical images, with different spectral and spatial characteristics, to portray the landslide features
that can be exploited for the visual detection and mapping of landslides. We maintain that the
results obtained in our test case are general, and should be considered for the optimal selection of
images for the detection and mapping event landslides.

## 2  The Assignano landslide

For our study, we selected the Assignano landslide, a slide-earthflow (Hutchinson, 1970) triggered
by intense rainfall in December 2013 in the northwest-facing slope of the Assignano village,
Umbria, central Italy (**Fig. 1**). The landslide develops in a crop area, where a layered sequence of
sand, silt and clay deposits crop out (Santangelo et al., 2015b). The slope failure is about 340 m
long, 40 m wide in the transportation area, and 60 m wide in the deposition area, and is
characterized by three distinct source areas, two located on the south-west side of the landslide and
third located on the north-east side of the landslide. The source and transportation area has an
overall length of about 230 m, and a width increasing from 10 to 40 m from the top of the source
area to the bottom of the transportation area. Elevation in the landslide ranges from 276 m along
the landslide crown, to 206 m at the lowest tip of the deposit. The source and transportation area is
bounded locally by sub-vertical, 2 to 4-m high escarpments. In the landslide, terrain slope averages
11°, and is steeper (12°) in the source and transportation area than in the deposition area (9°). The
landslide signature (Pike, 1988) is different in the different parts of the landslide. In the source and
transport area the signature is predominantly photographical (radiometric), whereas in the landslide
deposit it is mainly morphometric (topographic). The differences allow to separate the source and
transportation area from the deposition area.

## 3  Image acquisition

On 14 April 2014, we conducted an aerial survey of the Assignano landslide using a "X" shaped
frame octocopter with eight motors mounted on four arms (four sets of CW and CCW props) with
a payload capacity of around one kilogram, and a flight autonomy of about 20 minutes. The UAV
was equipped with a remotely controlled gimbal hosting a ©GoPro Hero 3 video camera and a
Canon EOS M camera. We controlled the flight of the UAV manually, relaying on the real-time
video stream provided by the ©GoPro. We kept the operational flight altitude of the UAV in the
range between 70 and 100 m above the ground. This allowed the Canon EOS M camera to capture
digital colour images of the landslide area with a ground resolution of about 2-4 cm, with the
single images having an overlap of about 70% and a side-lap of about 40%. For the accurate
geocoding of the images, we positioned 13 red-and-white, four-quadrants square targets,
20 cm $\times$ 20 cm in size, outside and inside the landslide. We obtained the geographical location
(latitude, longitude, elevation) of the 13 target centres using a Real Time Kinematic (RTK)
Differential Global Positioning System (DGPS), with a horizontal error of less than 3 cm. We
processed the 97 images using commercial, structure-from-motion software to obtain (i) a 3D point
cloud, (ii) a Digital Surface Model (DSM), and (iii) a digital, monoscopic, ultra-resolution (ground
sampling distance is $3 \times 3$ cm) ortho-rectified image in the visible spectral range, which we used
for the visual mapping of the Assignano landslide (**Table 1**).
To map the landslide, we also used a stereoscopic pair of VHR images taken on 14 April 2014 i.e.,
the same day of the UAV survey, by the WorldView-2 satellite that operates at an altitude of 496
km, and collects 46-cm panchromatic, and 1.84-m eight-band, multispectral (coastal blue, blue,
green, yellow, red, red edge, and near infrared-1, near-infrared-2) imagery at 11-bit dynamic range,
in the spectral range 0.400 – 1.040 μm. For the satellite imagery, the rational polynomial
coefficients (RPCs) are available, allowing for accurate photogrammetric processing of the images.
We used the RPCs to generate 3D models of the terrain from the stereoscopic image pair.
Exploiting the characteristics of the satellite image, we prepared four separate images for landslide
mapping, namely, (i) a monoscopic, "true colour" (TC) image, (ii) a monoscopic false-colour-
composite (FCC) image obtained from the composite near infrared, red and green (band 4,3,2), (iii)
a TC stereoscopic pair, and (iv) a FCC stereoscopic pair. We prepared separate maps of the
Assignano landslide through the visual interpretation of the four images (**Table 1**). Both satellite
and UAV images are free from deep shadows (**Fig. 2**).
To compare the images obtained by the UAV and the WorldView-2 satellite, we co-registered the
images, and we evaluated the co-registration on seven control points (**Fig. 3**), obtaining a Distance
Root Mean Square error, DRMS = 0.53 m, and a Circular Error Probability, $CEP_{50\%}$ = 0.42 m,
which we consider adequate for landslide mapping, and for the map comparison.

## 4  Landslide mapping

We prepared eight maps of the Assignano landslide using different approaches, images and
datasets, including two maps prepared through field surveys, four maps prepared through the visual
interpretation of monoscopic and stereoscopic satellite images, and two maps prepared through the
visual interpretation of the orthorectified images taken by the UAV (**Table 1**).
The field mapping and the image interpretation were carried out by independent geomorphologists.
The two geomorphologists who carried out the field activities i.e., the reconnaissance field mapping
and the RTK-DGPS survey, were not involved in the visual interpretation of the satellite and the
UAV images. Equally, the geomorphologist who interpreted visually the satellite and the UAV
images did not take part in the field activities. Visual interpretation of the remotely-sensed images
was performed by a single geomorphologist to avoid problems related to different interpretation
skills by different interpreters (Carrara et al., 1992). We then compared the eight resulting maps of
the Assignano landslide adopting a pairwise approach to quantify and evaluate the mapping
differences.
The geomorphologist who interpreted visually the images was shown first the 1.84-m resolution,
monoscopic satellite image, next the 1.84-m resolution stereoscopic satellite pair, and lastly the 3-
cm resolution UAV images. The monoscopic and the stereoscopic satellite images were first shown
in TC and then in FCC. Lastly, the interpreter was shown the draped ultra-resolution UAV image.
Selection of the sequence of the images given to the geomorphologist for the expert driven visual
interpretation was based on the assumption that for landslide mapping (i) the ultra-resolution
monoscopic images provide more information than the 1.84-m monoscopic or stereoscopic images,
(ii) for equal spatial resolution images, stereoscopic images provide more information than
monoscopic images, and (iii) for equal image type (monoscopic, stereoscopic), the FCC images
provide more information than the TC images. To prevent biases related to a possible previous
knowledge of the landslide, the interpreter was not shown the results of the reconnaissance field
mapping.

### 4.1  Field mapping

Field mapping of the Assignano landslide consisted in two synergic activities, (i) a reconnaissance
field survey, and (ii) a RTK DGPS aided survey. First, the reconnaissance field survey was
conducted by two geomorphologists (FF and MR) who observed the landslide and took
photographs of the slope failure from multiple viewpoints, close to and far from the landslide. The
geomorphologists draw in the field a preliminary map of the landslide exploiting the most recent
satellite image available at the time in Google Earth™, which was taken on 8 July 2013 i.e. (**Fig.**
**4**), before the landslide occurred. The reconnaissance field mapping was then refined in the
laboratory using the ground photographs taken in the field. We refer to this reconnaissance
representation of the Assignano landslide as "Map B".
Next, the same two geomorphologists (FF and MR) conducted an RTK DGPS aided survey
walking a Leica Geosystems GPS 1200 receiver along the landslide boundary, capturing 3D
geographic coordinates every about 5 m, in 3D distance. For the purpose, we used the SmartNet
ItalPoS real-time network service to transmit the correction signal from the GPS base station to the
GPS roving station. The estimated accuracy obtained for each survey point measured along the
landslide boundary was 2 to 5 cm, measured by the root mean square error (RMSE), on the ETRF-
2000 reference system. We refer to the cartographic representation of the Assignano landslide
produced by the RTK DGPS survey as "Map A". We consider this map as the "ground truth", and
we use it as a benchmark against which to compare the other maps. We acknowledge that mapping
a landslide by walking a GPS receiver around its boundary is an error prone operation e.g., because
in places the landslide boundary is not sharp, or clearly visible from the ground (Santangelo et al.,
2010). However, we maintain this is the most reasonable working assumption, and that the
geometrical information obtained by walking a GPS receiver along the landslide boundary was
superior to the information obtained through the reconnaissance field mapping (Map B)
(Santangelo et al., 2010).

### 4.2  Mapping through image interpretation

A trained geomorphologist (MS) used the three monoscopic images (i.e., the TC and FCC
monoscopic satellite images, and the monoscopic ultra-resolution UAV image) to perform a
heuristic, visual mapping of the Assignano landslide. For this purpose, the interpreter considered
the photographic (colour, tone, mottling, texture) and geometrical (shape, size, curvature, pattern
of individual terrain features, or sets of features) characteristics of the images (Antonini et al.,
1999). In this way, the geomorphologist prepared (i) "Map C" interpreting visually the
monoscopic, TC satellite image, (ii) "Map D" interpreting visually the monoscopic, FFC satellite
image, and (iii) "Map G" interpreting visually the monoscopic, TC UAV image (**Table 1**).
Next, the interpreter used the two stereoscopic satellite images (i.e., the TC and FCC images) to
prepare "Map E" and "Map F" (**Table 1**). In the stereoscopic images, the photographic and
morphological information is combined, favouring the recognition of the landslide features through
the joint analysis of photographic (colour, tone, mottling, texture), geometrical (shape, size, pattern
of features), and morphological terrain features (curvature, convexity, concavity). To analyse
visually the stereoscopic satellite images, the interpreter used the StereoMirror™ hardware
technology, combined with the ERDAS IMAGINE® and Leica Photogrammetry Suite (LPS)
software. To map the landslide features in real-world, 3D geographical coordinates, the interpreter
used a 3D floating cursor (Fiorucci et al., 2015).
To interpret the ultra-resolution UAV image, the interpreter overlaid ("draped") the image on
Google Earth™. For the purpose, we first treated the UAV image with gdal2tiles.py software to
obtain a set of image tiles compatible with the Google Earth™ terrain visualization platform. To
interpret visually the ultra-resolution UAV image, the interpreter overlaid ("draped") the image on
Google Earth™. For the purpose, we first treated the UAV image with the gdal2tiles.py software
to obtain a set of image tiles compatible with Google Earth™ terrain visualization platform. To the
best of our knowledge, the platform is the only free, 2.5D image visualisation environment that
allows the editing of vector (i.e., point, line, polygon) information. Other commercial (e.g.,
ArcScene) and open source (e.g., ParaView, GRASS GIS), 2.5D visualization tools do not provide
editing capabilities. Google Earth™ is a user-friendly solution for mapping single landslides, and
for preparing landslide event inventories for limited areas, with the possibility for the user to
visualize a landscape from virtually any viewpoint, facilitating landslide mapping. We refer to the
representation of the Assignano landslide obtained through the visual interpretation of the ultra-
resolution UAV image as "Map H".
For the visual interpretation of the satellite and the UAV images, the interpreter adopted a
visualization scale in the range from 1:1000 to 1:6000, depending on the image spatial resolution
(**Table 1**). The scale of observation was selected to obtain the best readability of each landslide
feature and the surroundings, which is a common practice in image visual analysis for landslide
mapping (Fiorucci et al., 2011). Hence, even if the maps were produced at slightly different
observation scales, the differences arising from the comparison are due to actual features (i.e., the
image resolution and radiometry), and not to the different observation scales.

## 231    5  Results

Using the described mapping methods, and the available satellite and UAV images (**Table 1**), we
prepared eight separate and independent cartographic representations of the Assignano landslide,
shown in **Fig. 5** as Map A to Map H.
Considering the entire landslide, visual inspection of **Fig. 5** reveals that the maps most similar to
the benchmark (Map A) are Map E, prepared examining the true colour (TC) stereoscopic satellite
image, and Map F, prepared examining the false colour composite (FCC) stereoscopic satellite
image. Conversely, the largest differences were observed for the landslide maps obtained through
the reconnaissance field survey (Map B), and the visual interpretation of the monoscopic satellite
images (Map C and Map D). Considering only the source and transportation areas (dark colours in
**Fig. 5**), interpretation of the UAV ultra-resolution images resulted in the landslide maps most
similar (Map G and Map H) to the benchmark (Map A). It is worth noticing the systematic lack in
the mapping of one of the two secondary landslide source areas located in the SW side of the
landslide, which was recognized only from the visual inspection of the ultra-resolution
orthorectified images taken by the UAV. In the field, this source area was characterized by small
cracks along the escarpment and a limited disruption of the meadow, making it particularly difficult
to detect and map. We argue that only the ultra-resolution images allowed for the detection of the
cracks. Considering only the landslide deposit (light colours in **Fig. 5**), the landslide mapping that
was more similar to the benchmark (Map A) was obtained interpreting the TC, stereoscopic
satellite images (Map E). We also note that in most of the maps the landslide deposit was mapped
larger (Map G, Map H) or much larger (Map B, Map C and Map D) than the benchmark (Map A).
**Table 2** lists geometric measures of the mapped landslides, including the planimetric measurement
of length, width and area (i) of the entire landslide, (ii) of the landslide source and transportation
area (dark colours in **Fig. 5**), and (iii) of the landslide deposit (light colours in **Fig. 5**). The length
and width measurements were obtained in a GIS as the length and the width of the minimum
oriented rectangle encompassing (i) the entire landslide, (ii) the landslide source and transportation
area, and (iii) the landslide deposit. Our benchmark (Map A) has a total area $A_L = 1.1 \times 10^4$ m$^2$, and
is $L_{LS} = 362$ m long and $W_{LS} = 71$ m wide. Amongst the other seven maps (Map B to Map H in
**Fig. 5**), the largest landslide is shown in Map B, obtained through the reconnaissance field
mapping, and has $A_L = 1.91 \times 10^4$ m$^2$, 71.1% larger than the benchmark. Conversely, the smallest
landslide is shown in Map F, with $A_L = 1.1 \times 10^4$ m$^2$, 4.6% smaller than the benchmark. The longest
and largest landslide is found in Map C, with $L_{LS} = 405$ m (11% longer than the benchmark) and
$W_{LS} = 113$ m (60% wider than the benchmark).
Considering the source and transportation area, in Map A (the benchmark) $A_{LS} = 5.4 \times 10^3$ m$^2$,
$L_{LS} = 228$ m, and $W_{LS} = 52$ m. The largest representation of the source and transportation area is
found in Map B (reconnaissance field mapping) with $A_{LS} = 7.4 \times 10^3$ m$^2$, 36.9% larger than the
benchmark, and the smallest source and transportation area is found in Map G, with
$A_{LS} = 5.2 \times 10^3$ m$^2$, 3.6% smaller than the benchmark. The longest source and transportation area is
found in Map F, with $L_{LS} = 239$ m, 5% longer than the benchmark, and the shortest source and
transportation area is shown in Map C, with $L_{LS} = 206$ m, 9.7% shorter than the benchmark. The
largest source and transportation area is shown in Map B, $W_{LS} = 60$ m, 15.7% wider than Map A,
and the narrowest source and transportation area is in Map C, $L_{LS} = 44$ m, 15.3% narrower than the
benchmark. Considering instead only the landslide deposit, our benchmark (Map A) has
$A_{LD} = 5.7 \times 10^3$ m$^2$, $L_{LS} = 153$ m, and $W_{LS} = 61$ m. The largest deposit is shown in Map B
(reconnaissance field mapping) and has $A_{LD} = 1.2 \times 10^4$ m$^2$, 103.4% larger than the benchmark,
whereas the smallest landslide deposit is shown in Map F, with $A_{LD} = 4.6 \times 10^3$ m$^2$, 19.8% smaller
than the benchmark. Analysis of the length and width of the landslide deposit reveals that Map C
shows the longest deposit, $L_{LS} = 206$ m, 35% longer than the benchmark, and Map H shows the
shortest deposit, $L_{LS} = 122$ m, 20.2% shorter than the benchmark. Similarly, the largest landslide
deposit is shown in Map C, $W_{LS} = 112$ m, 82.8% wider than the benchmark, and the narrowest
landslide deposit is portrayed in Map E, $W_{LS} = 56$ m, 8.2% less than the benchmark.
To compare quantitatively the different landslide maps, we use the error index $E$ proposed by
Carrara et al. (1992), adopting the pairwise comparison approach proposed by Santangelo et al.
(2015a). The index provides an estimate of the discrepancy (or similarity) between corresponding
polygons in two maps, and is defined as:

$$E = \frac{(A \cup B) - (A \cap B)}{(A \cup B)}; \ 0 \leq E \leq 1, \tag{1}$$

where, A and B are the areas of two corresponding polygons in the compared maps, and $\cup$ and $\cap$
are the geographical (geometric) union and intersection of the two polygons, respectively. $E$ spans
the range from 0 (perfect matching) to 1 (complete mismatch).
We compared the eight maps of the Assignano landslide (**Fig. 5**) adopting a pairwise approach,
and considering first only the landslide source and transportation area, next only the landslide
deposit, and lastly the entire landslide. **Fig. 6** summarizes the 84 values of the error index $E$, 28 for
the landslide source and transportation area (**Fig. 6 I**), 28 for the landslide deposit (**Fig. 6 II**), and
28 for the entire landslide (**Fig. 6 III**). On average, the source and transportation area exhibits
values of the error index smaller than the values found in the landslide deposit. This indicates that
in the source and transportation area the landslide maps are more similar than in the landslide
deposit. Inspection of **Fig. 6 I**, reveals a decrease of the error index in the source and transportation
area for the maps obtained interpreting the available images (from Map C to Map H), compared to
our benchmark obtained through the RTK DGPS survey ($0.15 \leq E \leq 0.38$), with Map G obtained
interpreting the TC, monoscopic, ultra-resolution UAV image. In the landslide deposit (**Fig. 6 II**),
the minimum difference ($E = 0.21$) was found comparing the benchmark to Map E, obtained
through the interpretation of the stereoscopic TC satellite image, and the largest difference
($E = 0.52$) was found comparing the benchmark to Map C, prepared interpreting the TC,
monoscopic, satellite image.
Comparison of the maps obtained through the interpretation of the monoscopic images (Map C and
Map D), and the maps obtained through the interpretation of stereoscopic (Map E and Map F) or
ultra-resolution images (Map G and Map H), reveals high values of the error index, which is
slightly worse in the landslide deposit. This is evident in the source and transportation area
($0.31 \leq E \leq 0.44$) (**Fig. 6 I**), and in the landslide deposit ($0.43 \leq E \leq 0.63$) (**Fig. 6 II**). Map C and
Map D are very similar, with a mapping error $E = 0.17$. Maps obtained through the interpretation
of stereoscopic satellite images (Map E and Map F, prepared using TC and FCC images,
respectively), and maps prepared by interpreting the UAV images (Map G and Map H), exhibit a
generally low value of $E$. In particular, $0.14 \leq E \leq 0.26$ in the landslide source and transportation
area, and $0.15 \leq E \leq 0.38$ in the landslide deposit. The reconnaissance field mapping (Map B)
exhibited the largest differences compared to all the other maps ($0.63 \leq E \leq 0.45$) in the landslide
source and transportation area, and $0.44 \leq E \leq 0.73$ in the landslide deposit. The large values of $E$
in the landslide deposit is probably due to lack of visibility of part of the landslide toe in the field.

## 6  Discussion

We discuss the ability of the different images used to detect and map the Assignano landslide (**Fig.**
**1**) to resolve the landslide photographical and morphological signatures, considering separately the
image spatial and spectral resolutions, and the image type i.e., monoscopic, stereoscopic, or
pseudo-stereoscopic. We treat each of the three factors separately, keeping the other two factors
constant. To evaluate the influence of the image spatial resolution on landslide mapping, we
compare to our benchmark (Map A) two true-colour (TC) monoscopic maps (Map C and Map G),
and two TC stereoscopic maps (Map E and Map H). Next, to evaluate the influence of the image
spectral resolution on the landslide mapping, we compare to the benchmark (Map A) the TC and
the false-colour-composite (FCC) monoscopic maps (Map C and Map D), and the corresponding
TC and FCC stereoscopic maps (Map E and Map F). Lastly, to assess the influence of the type of
image (i.e., monoscopic, stereoscopic, pseudo-stereoscopic) on the landslide mapping, we compare
to the benchmark (Map A) the monoscopic (Map C) and the stereoscopic (Map E) TC maps
(**Fig. 7A**), the two FCC maps (Map D and Map F) (**Fig. 7B**), and the maps obtained interpreting
the ultra-resolution images captured by the UAV (Map G and Map H). **Fig. 6** summarizes the
mapping errors $E$ obtained by the pairwise comparisons of the eight landslide maps shown in **Fig. 5**.
We first evaluate the role of the image spatial resolution in the production of the different maps of
the Assignano landslide. Inspection of **Fig. 6 I** reveals that the maps of the landslide source and
transportation area obtained from images characterized by the highest spatial resolution (i.e.,
Map G and Map H) exhibits the smallest errors ($E \leq 0.16$), when compared to the benchmark
(Map A). The mapping error obtained for Map C (TC, monoscopic, $E = 0.38$) is 2.5 times larger
than the error obtained using the ultra-resolution orhtorectified images taken by the UAV (Map G,
$E = 0.15$, and Map H, $E = 0.16$), whereas the error obtained from Map E (TC, stereoscopic,
$E = 0.23$) is smaller, and about 1.5 times larger than the error obtained for Map H (TC, pseudo-
stereoscopic, $E = 0.16$). In the landslide deposit (**Fig. 6 II**), the map obtained exploiting the
monoscopic, TC satellite image (Map C) exhibits an error $E = 0.52$, 1.7 times larger than the error
obtained using Map G (TC, monoscopic UAV, $E = 0.30$). Conversely, the error is smaller in the
map obtained from the 2-m spatial resolution, stereoscopic TC satellite image (Map E, $E = 0.21$)
than from the 3-cm spatial resolution, pseudo-stereoscopic image taken by the UAV (Map H,
$E = 0.30$). Collectively, the pairwise comparisons highlights an improvement of the quality of the
mapping of the landslide features that exhibits a distinct photographical signature, most visible in
the source and transportation area of the Assignano landslide, with an increase of the image spatial
resolution (**Fig. 6**). Use of the ultra-resolution image captured by the UAV did not result in an
improvement of the mapping in the deposition area of the Assignano landslide, where the landslide
exhibits a distinct morphological signature. We further observe that most of the landslide parts that
were not identified in the maps prepared using the satellite image are covered by vegetation, locally
bounded by small and thin cracks with an average width smaller than the size of the $2 \times 2$ m pixel.
In the satellite image, the cracks are located in pixels containing a mix of vegetation and bare soil,
making it difficult for the interpreter to recognize the cracks.
Next, we evaluate the effectiveness of the image spectral resolution, and for the purpose we
examine the mapping errors of Maps C and Map E (TC), and of Map D and Map F (FCC). The
mapping of the source and transportation area prepared using the false-colour-composite (FCC)
images (Map D and Map F) resulted in smaller errors than the mapping prepared using the
corresponding true-colour (TC) images (Map C and Map E), for both monoscopic and stereoscopic
images (**Fig. 6 I**). In the source and transportation area, the false-colour-composite emphasized the
presence or absence of the vegetation, and contributed locally to highlight the typical
photographical signature of the landslide, which helped the photo-interpreter to detect and map the
slope failure. Conversely, in the landslide deposition area (**Fig. 6 II**) use of the FCC images did not
result in a systematic reduction of the mapping error, when compared to the TC images. We
conclude that use of the additional information contributed by the Near Infrared (NIR) band in the
1.84-m resolution satellite image did not improve the quality of the mapping. On the other hand,
the contribution of the NIR in the 3-cm UAV image remains unknown.
Next, we evaluate the influence of the image type (i.e., monoscopic, stereoscopic, pseudo-
stereoscopic) on the mapping error by comparing (i) the TC images (Map C and Map E), (ii) the
FCC images (Map D and Map F), and (iii) the ultra-resolution UAV image (Map G and Map H).
Comparison of the TC, monoscopic (Map C) and stereoscopic (Map E) images revealed a mapping
error for the entire landslide $E = 0.48$, with the mismatch larger in the deposition area ($E = 0.59$)
than in the source and transpiration area ($E = 0.45$) (**Fig. 6**). A similar result was obtained
comparing the FCC, monoscopic (Map D) and stereoscopic (Map F) images, with a mapping error
for the entire landslide $E = 0.44$, and again the mismatch is larger in the deposition area ($E = 0.60$)
than in the source and transpiration area ($E = 0.36$). In the deposition area, where the morphological
signature of the Assignano landslide is strongest, the mapping error obtained comparing our
benchmark (Map A) to the landslide maps prepared using the monoscopic images (Map C and
Map D) is 2 times larger than the error observed for the maps prepared using the corresponding
stereoscopic images (Map E and Map F). The differences are smaller in the source and
transportation area, where the morphological signature of the landslide is less distinct. Direct
comparison of Map E (TC, stereoscopic) and Map F (FCC, stereoscopic) for the entire landslide
reveals a very small mapping error ($E = 0.15$), indicating the similarity of the two maps, which
were also very similar to the benchmark (Map A), $E \leq 0.20$.
Comparison for the entire landslide of the maps prepared using the ultra-resolution images captured
by the UAV (Map G and Map H) exhibits the smallest error of all the pairwise comparisons
($E = 0.08$) (**Fig. 6 III**), indicating the large degree of matching between the two maps. The degree
of matching is only marginally smaller in the source and transportation area, and in the deposition
area ($E = 0.15$). When compared to our benchmark (Map A), Map G and Map H exhibit a small
error ($E = 0.19$) for the entire landslide, which is larger in the deposition area ($E \leq 0.30$) and slightly
smaller in the source and transport area ($E \leq 0.15$). Interestingly, the mismatch with Map A (the
benchmark) is lower for the monoscopic (Map G) than for the pseudo-stereoscopic (Map H) map.
The finding highlights the lack of an advantage in using a pseudo-stereoscopic (2.5D) image for
mapping the Assignano landslide. We attribute this result to the low resolution of the (pre-event)
DEM used to drape the ultra-resolution image for visualization purposes, which did not add any
significant morphological information to the expert visual interpretation.
Joint analysis of **Fig. 5B** and **Fig. 6** reveals that, when compared to our benchmark (Map A), the
reconnaissance field mapping (Map B) exhibited the largest mapping error of all the performed
pairwise comparisons, with $E = 0.45$ in the source and transportation area, $E = 0.67$ in the landslide
deposit, and $E = 0.55$ for the entire landslide. We note than an error of $E = 0.50$ indicates that 50%
of the landslide area in one map (Map B, in this case) does not overlay with the other map (Map A,
the benchmark, in this case). Our results are similar to the results of tests performed to compare
field-based landslide maps against GPS-based surveys of single landslides (Santangelo et al.,
2010), the visual interpretation of very-high resolution stereoscopic satellite images (Ardizzone
et al., 2013), or the semi-automatic processing of monoscopic satellite images (Mondini et al.,
2013), and confirm the inherent difficulty in preparing accurate landslide maps in the field, unless
the mapping is supported by a GPS survey or a similar technology.
Our experiment showed that the mapping of the Assignano landslide obtained exploiting the ultra-
resolution images captured by the UAV (Map G and Map H) was comparable to the maps obtained
using the high resolution stereoscopic satellite image (Map E and Map F), and to the ground-based
RTK DGPS survey (Map A, the benchmark). We conclude that ultra-resolution images captured
by an UAV and the stereoscopic satellite images are well suited to map event landslides, at least in
physiographical settings similar to the one of our study area, and for landslides similar to the
Assignano landslide (**Fig. 1**).
For event landslide mapping, selection between ultra-resolution pseudo-stereoscopic UAV images
and very-high resolution stereoscopic satellite images depends on (i) the extent of the investigated
area, (ii) the available resources, including time and budget, and (iii) the accessibility to the study
area. The selection is largely independent from the landslide signature, at least for landslides similar
to the Assignano landslide. From an operational perspective, modern multi-rotor UAVs allow for
the acquisition of ultra-resolution images over small areas in a limited time, and at very low costs.
UAV-based surveys are flexible in their acquisition planning, and partly independent from the local
lighting conditions, including the cloud cover. As a drawback, UAVs are strongly (and negatively)
affected by wind speed and weather conditions, they allow for a limited flight time (currently
approximately 20 minutes in optimal conditions), which is reduced in bad weather conditions and
in cold environments, and typically have limited data storage capacity. Further, it must be possible
for the pilot to be at the same time near to the area to be surveyed and to maintain a safe distance
from the UAV, a condition that may be difficult to attain in remote or in mountain areas.
Collectively, the intrinsic advantages and limitations of modern UAVs make the technology
potentially well suited for the acquisition of ultra-resolution images for event, seasonal, and multi-
temporal mapping of single landslides, of multiple landslides in a single slope, or in a relatively
small area (a few hectares). The use of UAV images was recently proposed by Turner et al. (2015)
for determining the landslide dynamics, exploiting time series of images that can be constructed
using UAVs. The result is achievable thanks to centimetre co-registration accuracy of the UAV
images. Use of UAVs becomes impracticable with the increasing extent of the study area, largely
due to (i) the operational difficulty of flying UAVs over large areas (more than a few square
kilometres), and (ii) the acquisition and image processing time and associated cost, which increase
rapidly with the size of the study area (**Table 3**). On the other hand, very-high resolution,
stereoscopic satellite images have also advantages and limitations for the production of event,
seasonal and multi-temporal landslide inventory maps (Guzzetti et al., 2012). The main advantage
of the satellite images is that they cover large or very areas (tens to hundreds of square kilometres)
in a single frame with a sub-metre resolution well suited for landslide mapping through the expert
visual interpretation of the images (Ardizzone et al., 2013). On the other hand, limitations remain
due to distortions caused by different off-nadir angles in successive scenes, and to difficulties – in
places severe – to obtaining suitable (e.g., cloud-free) images at the required time intervals. This is
particularly problematic for the production of seasonal and multi-temporal landslide maps.
Information on the photographic or morphological signature of the typical, or most abundant,
landslides in an area, is important to selecting the optimal characteristics of the images best suited
for the production of an event, seasonal or multi-temporal landslide inventory map. Use of images
of non-optimal characteristics for a typical landslide signature in an area may condition the quality
(i.e., completeness, positional and thematic accuracy) of the landslide inventory. Where possible,
we recommend that the acquisition of images used for the production of event, seasonal or multi-
temporal landslide inventory maps is planned considering the typical landslide signature, in
addition to the purpose (event inventory, planning of monitoring systems), scale of the mapping
(i.e. regional or slope scale), and the size and complexity of the study area (**Table 3**).

## 7  Concluding remarks

We executed an experiment aimed at determining and measuring the effects of the image
characteristics on event landslide mapping. In the experiment, we compared landslide maps
obtained (i) through the expert visual interpretation of an ultra-resolution image taken by an UAV
with a ground resolution of $3 \times 3$ cm, and monoscopic and stereoscopic true-colour and false-
colour-composite (1.84 × 1.84 m) images taken by the WorldView-2 satellite, (ii) a reconnaissance
field survey of the landslide, and (iii) an accurate survey of the landslide obtained by walking a
GPS receiver along the landslide boundary. We conducted the experiment on a the Assignano
landslide (**Fig. 1**) triggered by intense rainfall in December 2013 in the northwest-facing slope of
the Assignano village, Umbria, central Italy. The landslide exhibited a predominant photographical
(radiometric) signature in the source and transport area, and a more distinct morphological
(topographic) signature in the deposition area. The results of our mapping experiment allow for the
following conclusions.
First, in the landslide source and transport area, where the signature of the slope failure was
primarily photographical (radiometric), mapping errors (Carrara et al., 1992; Santangelo et al.,
2015a) decreased with the increase of the spatial resolution of the images used for the expert visual
detection and mapping of the landslide. In the same area, the image photographic (radiometric)
characteristics (true-colour, false-colour-composite) and the image type (monoscopic,
stereoscopic) played a minor role in augmenting the quality of the landslide map. Conversely, in
the deposition area, where the signature of the landslide was primarily morphological
(topographical), mapping errors decreased using stereoscopic satellite images that allowed
detecting topographic features distinctive of the landslide.
FCC and TC in the stereoscopic satellite images give similar values of the error. This indicates that
the spectral resolution of the images does not provide useful information to recognize and map the
landslide morphological features. On the other hand, the high spatial resolution provided by the
UAV images reduces the error, when compared to the monoscopic satellite imagery. However, the
error obtained using the UAV images remains higher than that obtained using stereoscopic satellite
images, despite the latter having a pixel one order of magnitude larger than the UAV images. We
conclude that the increase in the spatial resolution improves the ability to map morphological
features when using monoscopic images.
Second, use of the stereoscopic satellite images resulted in more accurate landslide maps (lower
error index $E$) than the corresponding monoscopic images in the landslide deposition area, where
the signature of the landslide was primarily morphometric (topographic). This was expected, as the
stereoscopic vision allowed to better capture the 3D terrain features typical of a landslide (Pike,
1988), including curvature, convexity and concavity. Conversely, visual examination of the false-
colour-composite images resulted in more accurate maps than the corresponding true-colour
images in the landslide source and transport area, where the signature of the landslide was primarily
photographic (radiometric). This was also expected (Guzzetti et al., 2012). Expert visual
interpretation of pseudo-stereoscopic ultra-resolution image failed to provide better results than the
corresponding monoscopic ultra-resolution image, most probably because the DEM used to drape
(overlay) the image on the terrain information was of low resolution.
Third, the ultra-resolution ($3 \times 3$ cm) image captured by the photographic camera flown on-board
the Unmanned Aerial Vehicle (UAV) proved to be very effective to detect and map the landslide.
The expert visual interpretation of the monoscopic ultra-resolution image provided mapping results
comparable to those obtained using the about 2-m resolution, stereoscopic satellite image.
Fourth, a comparative analysis of the technological constrains and the costs of acquisition and
processing of ultra-resolution imagery taken by UAV, and of high, or very-high resolution imagery
taken by optical satellites, revealed that the ultra-resolution images are well suited to map single
event landslides, clusters of landslides in a single slope, or a few landslides in nearby slopes in a
small area (up to few square kilometres, Giordan et al., 2017) , and prove unsuited to cover large
and very large areas where the stereoscopic satellite images provide the most effective option
(Boccardo et al., 2015).
Fifth, our field-based reconnaissance mapping (Map B) provided the least accurate mapping
results, measured by the largest mapping error ($E = 0.55$ for the entire landslide) when compared
to the benchmark map (**Fig. 6**). Our results confirm the inherent difficulty in preparing accurate
landslide maps in the field through a reconnaissance mapping (Santangelo et al., 2010).
Although we conducted our study on a single landslide (**Fig. 1**), we maintain that the findings are
general, and can be useful to decide on the optimal imagery and technique to be used when planning
the production of a landslide inventory map. We emphasize that the technique and imagery used
to prepare landslide inventory maps should be selected depending on multiple factors, including (i)
the typical or predominant landslide signature (photographic or morphological), (ii) the scale and
size of the study area (a single slope, a small catchment, a large region), and (iii) the scope of the
mapping (event, seasonal, multi-temporal, Guzzetti et al., 2012).

## 8 Acknwoledgements

FF and MS were supported by a grant of Italian Dipartimento della Protezione Civile. We thank Andrea Bernini and Mario Truffa, Servizio Protezione Civile della Città Metropolitana di Torino, for flying the UAV over the Assignano landslide.

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

**Table 1.** Characteristics of the images used to identify and map the Assignano landslide (**Fig. 2**).
O: order in the sequence of images shown to the interpreter. Platform used to capture the image:
W, WorldView-2 satellite; U, UAV. Resolution (ground resolution), in metre. Spectral (image
spectral composite): TCC, True Colour Composite (Red, Green, Blue); FCC, False Colour
Composite (Near infrared, Red, Green). Type (image type): M, monoscopic; S, stereoscopic; P,
pseudo-stereoscopic. Map: Corresponding landslide map (**Fig. 5**).

| O | Platform | Resolution | Spectral | Type | Map |
|---|---|---|---|---|---|
| 1 | W | 1.84 | TC | M | C |
| 2 | W | 1.84 | FCC | M | D |
| 3 | W | 1.84 | TC | S | E |
| 4 | W | 1.84 | FCC | S | F |
| 5 | U | 0.03 | TC | M | G |
| 6 | U | 0.03 | TC | P | H |



**Table 2**. Comparison of the total landslide area ($A_L$), the landslide source and transportation area
($A_{LS}$), the landslide deposit ($A_{LD}$), the width and length of the entire landslide ($W_L$, $L_L$), of the
source and transportation area ($W_{LS}$, $L_{LS}$), and of the deposit ($W_{LD}$, $L_{LD}$), for eight separate and
independent cartographic representations of the Assignano landslide. EL, entire landslide; ST,
landslide source and transport area; LD, landside deposit. See **Table 3** for the characteristics of the
single maps.

| | | Map A | Map B | Map C | Map D | Map E | Map F | Map G | Map H |
|---|---|---|---|---|---|---|---|---|---|
| Landslide area ($m^2$) | | | | | | | | | |
| EL | $A_L$ | $1.11\times10^4$ | $1.91\times10^4$ | $1.53\times10^4$ | $1.52\times10^4$ | $1.09\times10^4$ | $1.06\times10^4$ | $1.19\times10^4$ | $1.16\times10^4$ |
| ST | $A_{LS}$ | $5.40\times10^3$ | $7.40\times10^3$ | $3.64\times10^3$ | $4.02\times10^3$ | $5.71\times10^3$ | $6.03\times10^3$ | $5.21\times10^3$ | $5.70\times10^3$ |
| LD | $A_{LD}$ | $5.73\times10^3$ | $1.17\times10^4$ | $1.16\times10^4$ | $1.12\times10^4$ | $5.15\times10^3$ | $4.59\times10^3$ | $6.70\times10^3$ | $5.87\times10^3$ |
| Landslide length (m) and width (m) | | | | | | | | | |
| EL | $W_L$ | 70.7 | 97.8 | 113.4 | 109.9 | 61.4 | 61.25 | 89.9 | 85.3 |
| | $L_L$ | 362.0 | 387.5 | 404.7 | 391.2 | 354.6 | 359.5 | 343.3 | 349.1 |
| ST | $W_{LS}$ | 51.5 | 59.6 | 43.6 | 49.2 | 51.92 | 54.3 | 49.5 | 50.5 |
| | $L_{LS}$ | 227.9 | 229.7 | 205.9 | 208.0 | 239.0 | 239.2 | 234.7 | 237.3 |
| LD | $W_{LD}$ | 61.0 | 98.69 | 111.5 | 109.0 | 56.0 | 57.6 | 89.9 | 81.9 |
| | $L_{LD}$ | 152.7 | 172.1 | 206.2 | 203.5 | 129.8 | 134.7 | 139 | 121.8 |



**Table 3**. Comparison of the estimated cost, acquisition and pre-processing time, and storage
requirement for an area of 4 km$^2$ (2 km × 2 km) and for an area of 100 km$^2$ (10 km × 10 km), for
monoscopic and stereoscopic satellite images, and for an area of 15 km$^2$ for photographic images
captured by an UAV.

| | Satellite monoscopic | | Satellite stereoscopic | | UAV | |
|---|---|---|---|---|---|---|
| | 4 km$^2$ | 100 km$^2$ | 4 km$^2$ | 100 km$^2$ | 4 km$^2$ | 15 km$^2$ |
| Acquisition cost (€) | 1.500 | 1.500 | 3.500 | 3.500 | 1.000 | 3.000 |
| Pre-processing cost (€) | 50 | 50 | 50 | 50 | 250-300 | 3.000 |
| Acquisition time (day/person) | 7-60 | 7-60 | 7-60 | 7-60 | 1 | 4 |
| Pre-processing time (hr/person) | 1 | 1 | 1 | 1 | 5-6 | 20-24 |
| Storage (GB) | 0.5 | 0.5 | 1 | 1 | 12 | 50 |
| Resolution (m) | 2 | 2 | 2 | 2 | 0.02 | 0.02 |
| | | | | | | |
| Morphologic signature | no | no | yes | yes | yes | yes |
| Photographic signature | yes | yes | yes | yes | yes | yes |




## Figure captions

**Figure 1**. The Assignano landslide, located near Collazzone, Umbria, central Italy. (A) global view of the landslide. (B) detail of the landslide source area. (C) detail of the landslide transportation area. (D) detail of the landslide deposit. Base image obtained overlaying ("draping") the image on Google Earth™. Red line is the boundary of the landslide obtained using the RTK DGPS (benchmark).

**Figure 2**. Images used to map the Assignano landslide. (A) TC WordView-2 satellite image, (A-I) detail of the source area and (A-II) detail of the landslide deposit. (B) WordView-2 satellite image in FCC, (B-I) detail of the source area and (B-II) detail of the landslide deposit. (C) UAV monoscopic image and C-I a detail of the source area and C-II a detail of the deposition area.

**Figure 3**. Position of the seven GCPs used to evaluate the co-registration of WordView-2 satellite image (A) and UAV image (B). Corresponding points are illustrated with the same symbol. Differences of the coordinates of the corresponding points along X (i.e., E-W direction, ΔX) and along Y (i.e., N-S direction, ΔY) are provided in metres on the left of the figure.

**Figure 4**. (A) Overview of the Assignano landslide area in Google Earth™ taken on 8 July 2013. Photo shooting points and photograph taken (B) close to the landslide and (C) from a viewpoint. The photographs taken in the field and the Google Earth™ image were used to prepare the reconnaissance field map.

**Figure 5**. Eight independent cartographic representations of the Assignano landslide, "Map A" to "Map H". Map A obtained through a RTK DGPS survey is considered the "benchmark", and shown as a thick black line in the other maps. Map B obtained through reconnaissance field mapping. Map C to Map F obtained through the expert visual interpretation of the satellite images. Map G and Map H obtained through the expert visual interpretation of the orthorectified image taken by the UAV. See **Table 1** for image characteristics. Dark colours show the landslide source and transportation area. Visual inspection of the images reveals the maps most similar to the benchmark.

**Figure 6**. The error index ($E$) proposed by Carrara et al. (1992), was used to compare quantitatively the different landslide maps. (I) Error index matrix for the landslide source and transportation area. (II) Error index matrix for the landslide deposit. (III) Error matrix for the entire landslide. $E$ spans

the range from 0 (perfect matching) to 1 (complete mismatch).
**Figure 7**. Comparison of landslide maps prepared for the Assignano landslide, Umbria, Central
Italy. (A) Landslide map obtained from a monoscopic (Map C, dark yellow line) and a stereoscopic
(Map E, light blue line), true-colour (TC) WordView-2 satellite image (base image), and a mapping
of the landslide obtained by walking a GPS receiver along the landslide boundary (Map A, black
line). (B) Landslide map obtained from a monoscopic (Map D, yellow line) and a stereoscopic
(Map F, cyan line), false-colour-composite (FCC) WordView-2 satellite image, and a mapping
obtained by walking a GPS receiver along the landslide boundary (Map A, black line). (C)
Landslide map obtained from field survey (Map B, pink line) and from a monoscopic, TC, ultra-
resolution image captured by an UAV (Map G, purple line), and the mapping obtained by walking
a GPS receiver along the landslide boundary (Map A, black line).

**Figure 1**

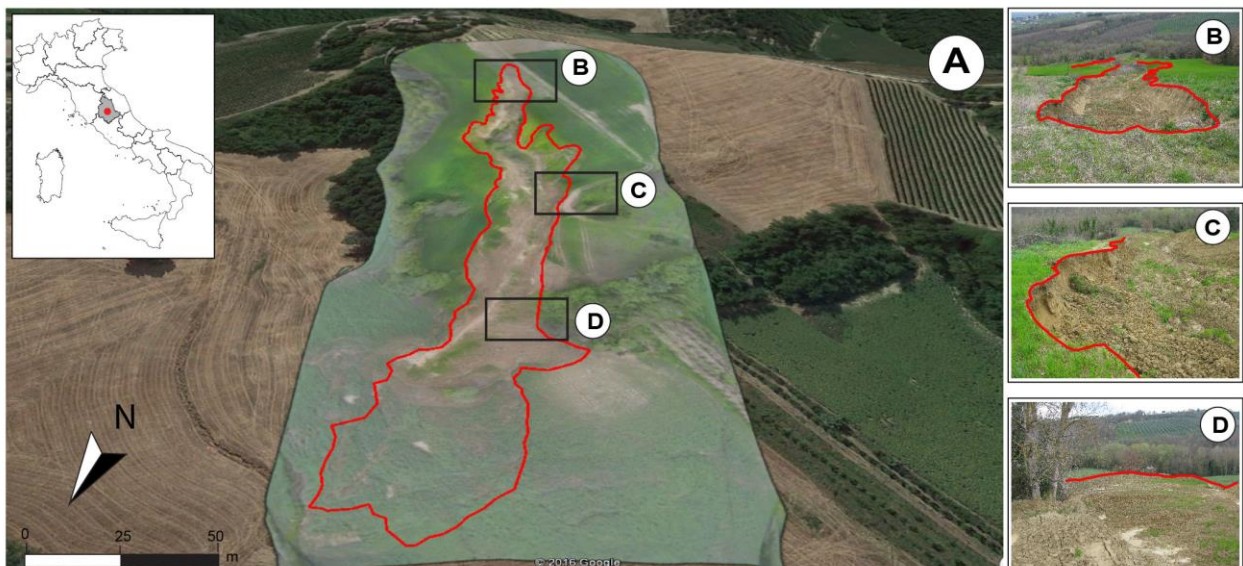



**Figure 2**

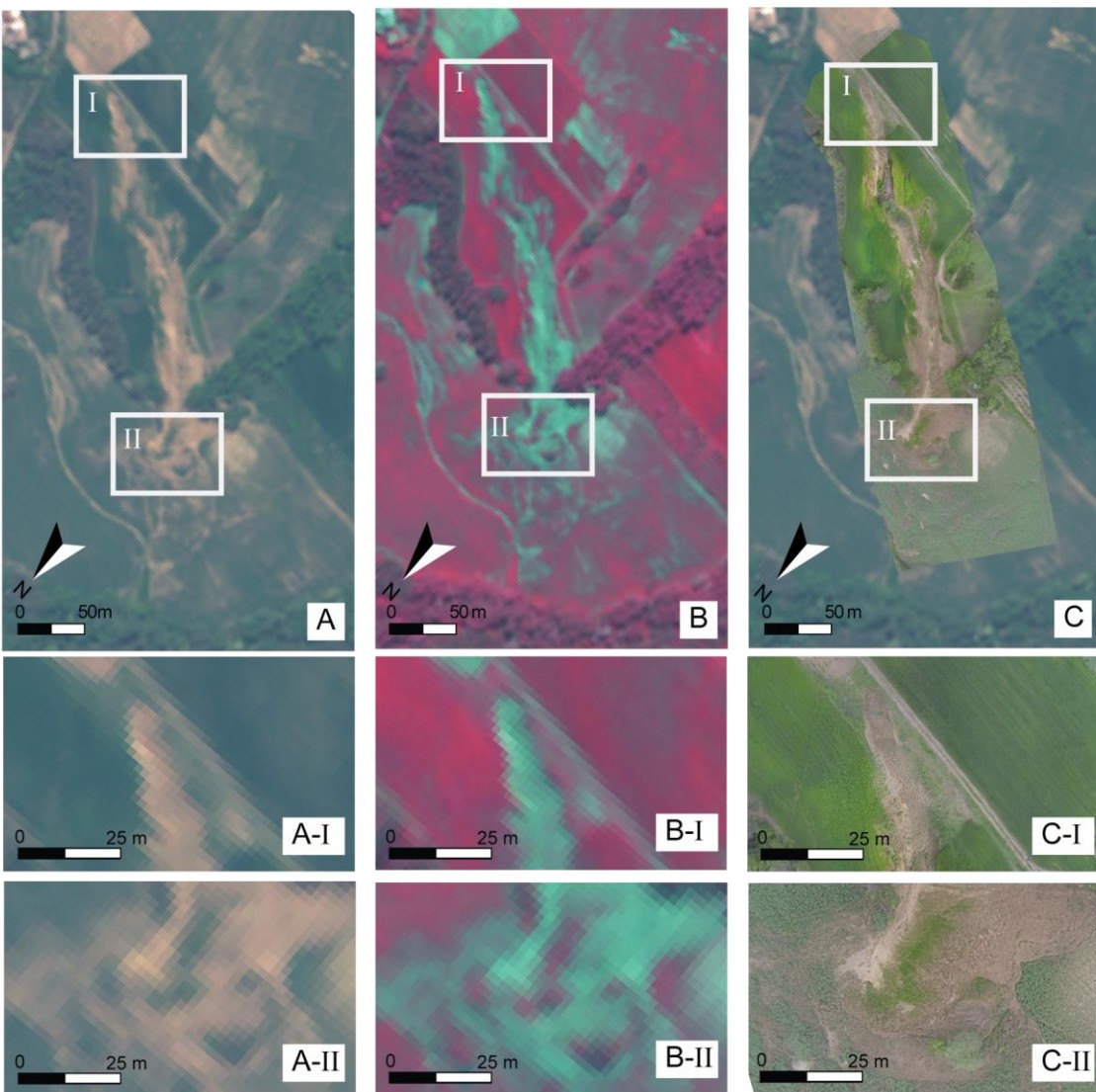


**Figure 3**

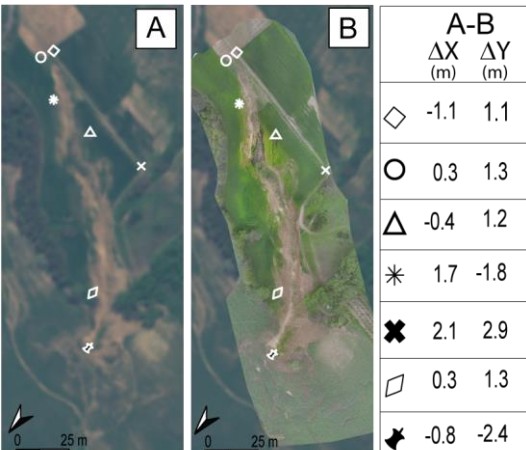


**Figure 4**

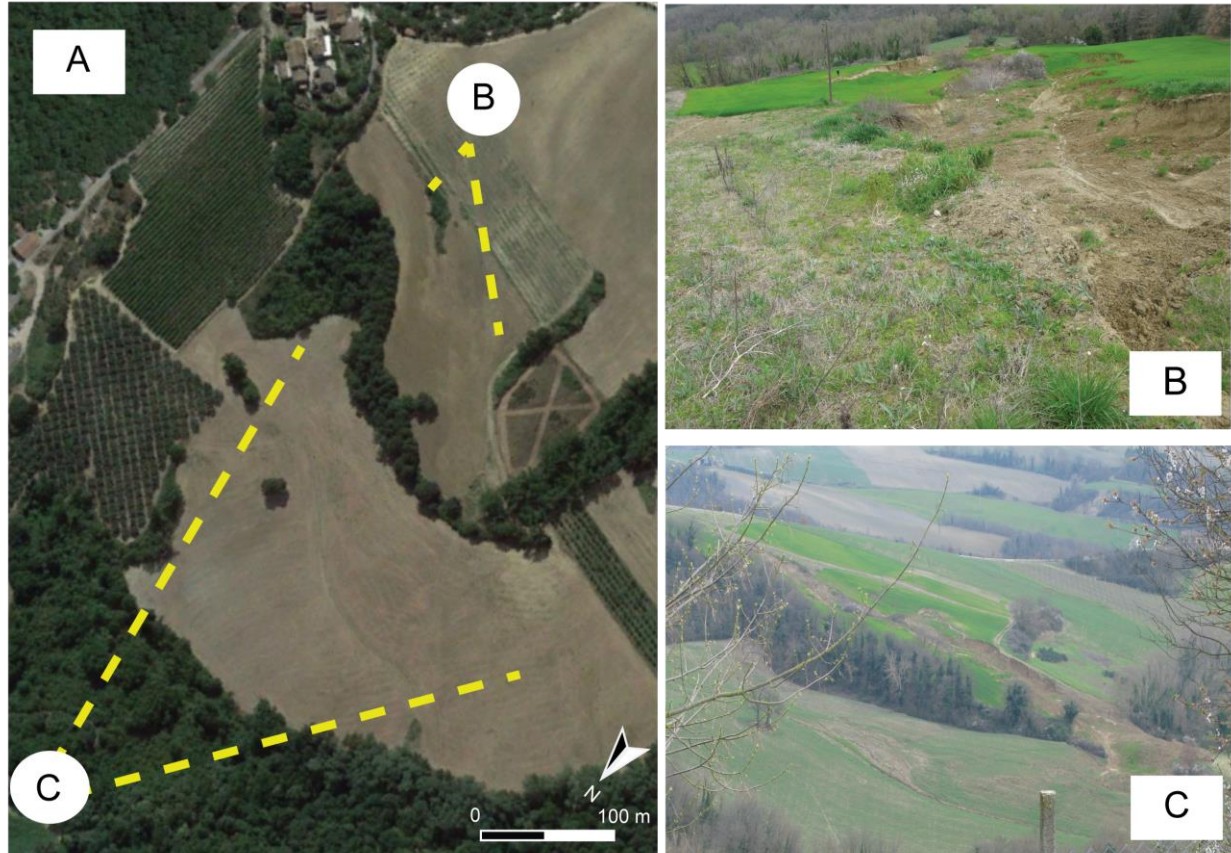


**Figure 5**

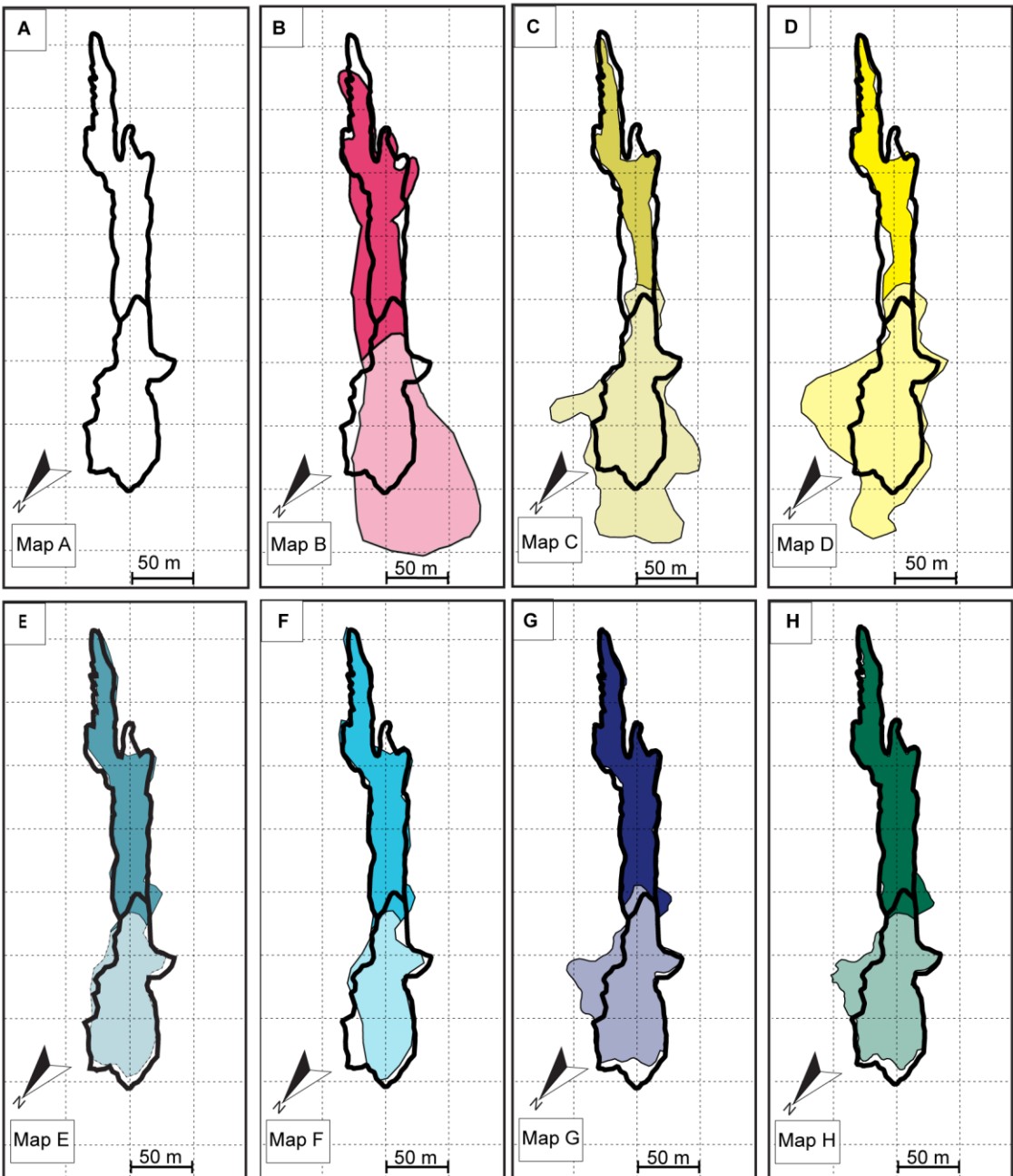


**Figure 6**

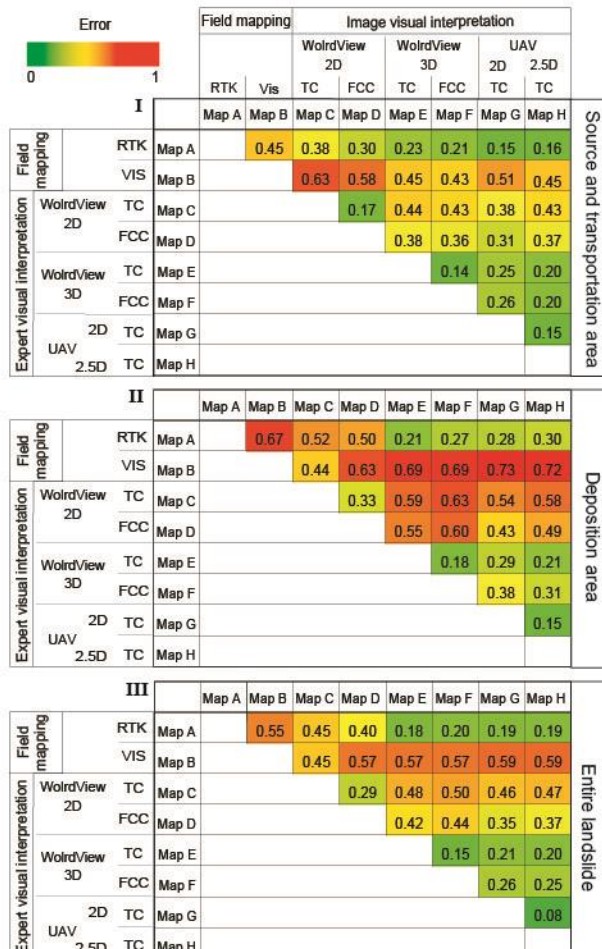




**Figure 7**

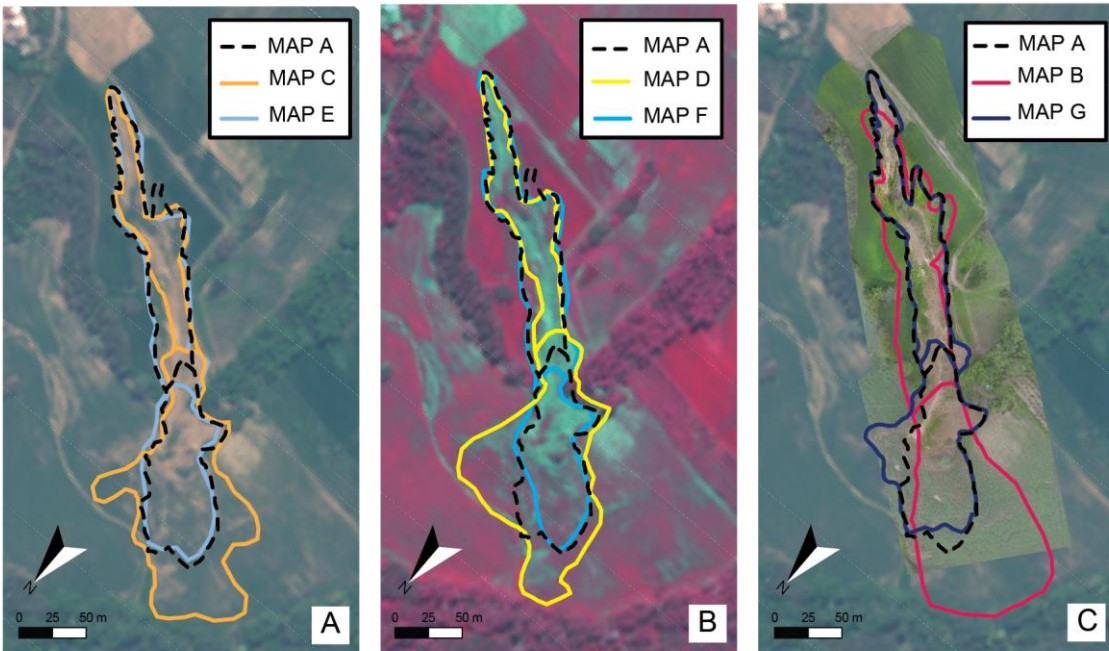
