# Peer review of "Criteria for the optimal selection of remote sensing images to map event landslides"

_Natural Hazards and Earth System Sciences, 2017_

## Referee Comment (RC1) · Anonymous Referee #1 · 15 May 2017

The authors present a study focussing on an expert-based interpretation of imagery data with the aim of mapping landslide features of a single landslide. Various data are tested and the mapping results are compared to reference data and field mappings. The authors then give recommendations regarding the feasibility of the different mapping techniques and imagery data for landslide mapping.

The employed methods are standard methods (dGPS, heuristic landslide mapping techniques), so there is no methodical innovation. The used software are commercial products. The results are difficult to reproduce, since only one expert did all the mapping. It would be interesting to see, how the landslide would have been mapped by further experts (>10). Furthermore, it remains unclear if the results are transferable to the relevant scale of event landslide inventories or to other types of landslides.

[Figure]

The text is generally well written, but there are some minor mistakes of grammar and style. Some of them are addressed below, but it would be out of scope to raise every issue. Therefore I recommend careful copy editing. Below, I focus on issues concerning the scientific content of the manuscript. Where numbers are given in the specific comments they refer to the manuscript page and line. A major revision carefully addressing the raised issues below is required, before the paper can be considered for publication.

Specific comments:

Throughout the manuscript: Unless you conducted statistical hypothesis tests, avoid the term 'significant' or 'significantly' and replace it by other attributes (e.g. distinct).

Section 1

Consider introducing the principle of heuristic, visual mapping of landslide features based on the interpretation of landslide signs ('geometric signature'; Pike, 1988). This is well explained in Section 4.2. However, an introductory description of the procedure would benefit the understanding of the reader. In this context, also explain the advantage of stereoscopic over monoscopic interpretation techniques.

Comment on the positional accuracy of landslide mapping, which depends on the data it is based on. UAV-based imagery theoretically allows for mapping with sub-decimetre details and accuracy. But is that even necessary? On the contrary, landslide mapping based on satellite imagery will result in less detailed features with less positional accuracy. Consider addressing the necessary positional accuracy of the mapped landslide features with respect to the intended use/scale of the compiled landslide inventory. For an overview map of landslides triggered by a specific event the positional accuracy will not play a great role. If the inventory should serve as input data for landslide modelling however, the positional accuracy of the mapped landslides is of great importance.
Consider adding a sentence addressing the potential of UAV-based imagery for efficiently analysing changes over time (e.g. Turner et al., 2015).

Section 2

Describe the landslide in much more detail. Which type of landslide is it, what type of material is involved (since the area seems to be not very steep) and what are the causes and failure mechanism? Have there been changes during the winter months (e.g. retrogressive failure, erosion)? Is the area cultivated/what is the land cover/land use?

Section 4

Add a figure with examples of each imagery data considered for landslide mapping showing each time (without the mapped features) (i) the whole landslide and (ii) a detailed view (e.g. one of the areas specified in Fig. 1). Also add the map used for the field mappings and the positions of the seven ground control points.

Add a table specifying what was done by whom and when. Also include the abbreviations of the persons.

When were the DGPS measurements conducted? Was the landslide deposition area clearly discernible in the field? If not, this data cannot be used as reference for the other datasets. Were there any changes in the period between the triggering of the landslide and the acquisition of the UAV- and satellite-images (e.g. vegetation ingrowth)?

Describe the 2.5D pseudo-stereoscopic data in more detail. Why was the landslide mapping based on the orthorectified UAV-imagery done in Google Earth and not using a more suited GIS software? Did you use the DEM included in Google Earth for aiding the mapping procedure? Why didn't you consider a DEM based on the UAV-point cloud? Since in most of the scene there is no high vegetation (trees), the landslide's morphology should be represented well. Also other derivatives of the resulting UAV-
based DEM (e.g. shaded reliefs, e.g. Niethammer et al., 2010) could be used for landslide mapping. Then, also the morphometric features could have been mapped better using the UAV data.

Describe the transfer of mapped landslide features from Google Earth to the GIS. Which GIS software was used? Which coordinate system/projection was used for the individual datasets (can Google Earth handle ETRF-2000)?

Mention that you mapped the source/transportation area and the deposition area as separate landslide features. How did you discern the source/transportation area from the deposition area? Are there indicators beyond subjective visual recognition? How did you treat shadows during landslide mapping?

Section 5

Are the length, width and area measurements planimetric (projected) measures?

Section 6

Comment on the comparability of landslide features mapped on different scales (1:1.000 to 1:6.000).

P10, L10-14: This statement only applies for the used data (UAV-based orthorectified image supported by the DEM of Google Earth). If another DEM was used (e.g. based on the point cloud derived from the UAV-data), the mapping results may improve.

Technical comments:

Abstract:

L18: change Goggle to Google

Do not use abbreviations (GPS, GIS)

P1, L7: change to: through field surveys. . .

P2, L26: explain VHR here (now on P3, L29)

P3, L3-L6: Consistently use either abbreviations or the written out form of the slope direction.

P3, L19: change to: ©GoPro camera.

P3, L31: consistently specify the spatial resolution of the satellite imagery data (either 1.84 or 1.85 m) throughout the paper, also in Table 1.

P4, L1: change to: For the satellite imagery,. . .

P4, L5: mention that 432 are the considered bands

P4, L14: in case of the UAV-based imagery, landslide mapping is based on the generated orthorectified image, right? Change to: . . .visual interpretation of the orthorectified image derived from the images taken by the UAV (or something similar). Also on P6, L12 and the caption of Fig. 2.

P4, L27: Was the satellite imagery resampled to 2 m? Otherwise state the original spatial resolution of 1.84 m (or 1.85 m).

P5, L6: change to: before the landslide occurred.

P5, L7: change filed to field

P5, L13: use RMSE as abbreviation

P5, L24: change to: For this purpose,. . .

P7, L32: change (0.38<E<0.15) to (0.38≥E≥0.15)

P8, L7-8: there is a verb missing in the sentence. E.g. begin with 'This is particularly true for. . .'

P9, L30: change to: the mismatch is. . .

P11, L3: change to: and typically have a limited. . .

P11, L12: change to: large areas

P11, L13: change to: sub-metre

P11, L18: change to: morphological signature

P11, L19: change to: selecting

P11, L21: what do you mean by geographic accuracy? Positional accuracy?

Figures and Tables

Figure 1: add information on the shown datasets in Fig. 1A (also add a reference to Google Earth), also specifying the source of the polygons and -lines.

Figure 2: Add a north arrow. Change DGPC to DGPS in the caption.

Table 1: change meter to metre in the caption

Consider the following papers:

Niethammer, U., S. Rothmund, M. R. James, J. Travelletti, and M. Joswig 2010: UAV-based remote sensing of landslides, Int. Arch. Photogram. Remote Sensing Spatial Info. Sci., 38(5), 496-501.

Petschko, H.; Bell, R. Glade, T. 2016: Effectiveness of visually analyzing LiDAR DTM derivatives for earth and debris slide inventory mapping for statistical susceptibility modeling, Landslides 13(5), 857-872.

Turner, D.; Lucieer, A. de Jong, S. M. 2015: Time Series Analysis of Landslide Dynamics Using an Unmanned Aerial Vehicle (UAV), Remote Sensing 7(2), 1736-1757.

---

## Referee Comment (RC2) · Anonymous Referee #2 · 24 May 2017

This paper aims at comparing different geological mapping of the perimeter of an italian landslide within a temperate area partially covered with forested vegetation. The authors realize that high resolution, various wave length and stereoscopic views helps a lot in order to precise the external geometry of some sections of this landslide (crown-transport and sedimentation areas). Moreover authors quantify the misfit in between those different mappings relative to a benchmark (Field RTK DGPS survey) through a useful error matrix. The differences in the mapping partly derived from the forest cover that hide the exact perimeter of this landslide.

To my point of view the main teaching of this paper is not new as geologists/geomorphologists experts in mapping know since a long time that very high resolution, as well as False color composition (relative to True color) and stereoscopic analyses are major and compulsory keys for a precise and exact geologi-

cal/geomorphological mapping of any geological/geomorpholoigcal objects. Moreover, planimetric differences of mapped objects also are not new see for instance the work on various fractal distances on the measurements of a Britany shoreline that change a lot function of the scale and the resolution (see the basic work of the mathematician benoit Mandelbrot ENSMP Fontainebleau and his team in the 1980's).

The interest of this paper is to illustrate it correctly with a pedagogic example and to recall to any scientists these facts using a specific example. In that sense it is interesting for NHESS to publish it.

Anyway, I propose major revision for this paper and to make some more work in order to optimize the inputs of this paper : 1. Could you differenciate more clearly the 3 sections of this landslide on those various mapping erosionnal part (crown), transport section, and at least the sedimentationnal section (toe). With which image (and why) do we have the best and the more exact geological mapping of this landslide ? 2. Could you precise the inputs and differences through local case examples on a new figure of high resolution DTM, FCC and stereoscopic mapping in order that the reader will be able to get an independant position. 3. Please finally dealing with your experience on that landslide what (and why) is your best and more exact mapping ? please justify it ? 4. What is your best methodological solution to map precisely such italian landslides ? 5. If you compare the benchmark and the mappings the map E (stereoscopic image seems the best fit... could you comment on that ? 6. Definitely I do not understand the misfit between map A (field DGPS survey) and map B (field landslide mapping), could you comment on the experts landslide mapping discrepancies ?

into details: p4. line92-94: precise ...predominantly photogrammetric... and morpho-metric... p5. line 108 : an horizontal... page 6, line 162: field page7, line 182: perform an heuristic page 8, l221: this source area was characterized by small cracks (please show on a figure those features. page 9, line 228 to 257 the comments of the table 2 is difficult to follow could you find an easier solution more convenient and easier to under-stand to present those results ? P.11 , line 282 poor agreement please precise... P.11,

line 287: good agreement please precise... P.13, line 343 please precise a sentence on the resolution of the NIR datasets used herein and what could be the inputs if the NIR dataset if it would have 3x3cm2 ground resolution... P.14 line 385 ...is comparable... is to my point of view poor... We do need to have precision on the differences in between mapping from stereoscopic and high resolution... You are working on a local case example you should go farther on your reflection and give to the scientific community your choice of the best way to map such kind of landslide. P.14, lines 396-397: and partly independant from the local lighting conditions including the cloud cover... please precise... p.15, l 407 flying P.15, l.412 : large or very large areas... p.16, l 447-448: a better resolution and spectral resolution did not contribute significantly to reducing the mapping errors : ??? please precise... p.16, l461: prove to be very effective P. 17, 8 l 481 acknowledgments the references needs to be carefully checked. Fig.1 to 4: please give comments within the legend that give the key points of the figures. Add a figure with specific details inputs of the landslide and compare it to the different geoloigcal mappings.
* * *

---

## Author Comment (AC1) · 7 Jul 2017

Answer to referee # 1

RC - Referee comments

AC- Author comments

RC-The authors present a study focussing on an expert-based interpretation of imagery data with the aim of mapping landslide features of a single landslide. Various data are tested and the mapping results are compared to reference data and field mappings. The authors then give recommendations regarding the feasibility of the different mapping techniques and imagery data for landslide mapping. The employed methods are standard methods (dGPS, heuristic landslide mapping techniques), so there is no

methodical innovation. The used software are commercial products. The results are difficult to reproduce, since only one expert did all the mapping. It would be interesting to see, how the landslide would have been mapped by further experts (>10). Furthermore, it remains unclear if the results are transferable to the relevant scale of event landslide inventories or to other types of landslides.

AC-We thank this Reviewer (R1) for this comment. As correctly noted by R1, in this work we adopted different (standard) techniques and digital images to produce a landslide inventory. The techniques consist in field mapping and photointerpretation. For the latter we used six different digital products. However, we point out that the aim of the effort was not to investigate the feasibility of techniques, nor to give absolute criteria to choose among different images. The study focuses on the definition of criteria for the selection of remote sensing images for the specific purpose of mapping event landslides. For this reason, we relied upon a single expert to perform the landslide recognition and mapping. We considered the possibility to use more experts. However, this would have added the uncertainty inherent in the subjective interpretation of aerial photography for landslide mapping (see e.g., Carrara et al., 1992, Uncertainty in evaluating landslide hazard and risk. ITC Journal, 172–183). The uncertainty inherent with the interpreters would have mixed (and covered partially) the "signal" from the different imagery used for our experiment. Since the scope of the research was to investigate the information content of the imagery (and not of the interpreters) we ruled out the possibility of using more interpreters. Further, the researcher geomorphologist who interpreted the images and prepared the maps (MS) has a significant experience in photointerpretation for landslide mapping (he has prepared 25 landslides maps, including event maps, geomorphological maps, multi-temporal maps, covering more than 4000 km2, obtained using both monoscopic and stereoscopic satellite images and stereoscopic aerial photographs). Thanks to the expertise of the mapper, in each digital image the relevant features of the landslide were recognized fully. Thus, we are confident that differences among the six maps are to be ascribed to the sole resolving power of the different images. We have clarified this point in the text (see below). Moreover, we selected a landslide having both morphological and photographical signatures, which are the two key features that allows to recognize and map landslides from digital images. For this reason, we maintain that the results we have obtained are valid at all scales, and for most landslide types.

RC-The text is generally well written, but there are some minor mistakes of grammar and style. Some of them are addressed below, but it would be out of scope to raise every issue. Therefore, I recommend careful copy editing. Below, I focus on issues concerning the scientific content of the manuscript. Where numbers are given in the specific comments they refer to the manuscript page and line. A major revision carefully addressing the raised issues below is required, before the paper can be considered for publication.

AC-We thank R1 for reading carefully or Manuscript. We amended the text following R1 suggestions, where applicable.

Specific Comments

RC-Consider introducing the principle of heuristic, visual mapping of landslide features based on the interpretation of landslide signs ('geometric signature'; Pike, 1988). This is well explained in Section 4.2. However, an introductory description of the procedure would benefit the understanding of the reader. In this context, also explain the advantage of stereoscopic over monoscopic interpretation techniques.

AC-In the Introduction, we added the following language to clarify the text:

"The heuristic visual mapping of landslide features is based on the systematic analysis of image photographic and morphological characteristics such as colour, tone, mottling, texture, shape, size, curvature (Pike 1988). These photographic and morphological characteristics encompasses all the possible landslide features that can be used for the (visual) interpretation of the available imagery."

RC-Consider addressing the necessary positional accuracy of the mapped landslide

features with respect to the intended use/scale of the compiled landslide inventory.

AC-We accepted this suggestion of R1. In the Discussion section, we added the following sentence: "Where possible, we recommend that the acquisition of images used for the production of event, seasonal or multi-temporal landslide inventory maps is planned considering the typical landslide signature, in addition to the purpose (event inventory, planning of monitoring systems), scale of the mapping (i.e. regional or slope scale), and the size and complexity of the study area (see Table 3)."

RC-Consider adding a sentence addressing the potential of UAV-based imagery for efficiently analysing changes over time (e.g. Turner et al., 2015). Added to Discussion section.

AC-We accepted this suggestion of R1. In the Discussion, we added the following sentence: "The use of UAV images was recently proposed by Turner et al. (2015) for determining the landslide dynamics, exploiting time series of images that can be constructed using UAVs. The result is achievable thanks to centimetre co-registration accuracy of the UAV images."

RC-Which type of landslide is it, what type of material is involved (since the area seems to be not very steep) and what are the causes and failure mechanism?

AC-We accepted the comment of R1, and changed the text as follows: "For our study, we selected the Assignano landslide, a slide-earthflow (Hutchinson, 1970) triggered by intense rainfall in December 2013 in the northwest-facing slope of the Assignano village, Umbria, central Italy (Fig. 1). The landslide develops in a crop area, where a layered sequence of sand, silt and clay deposits crop out (Santangelo et al., 2015)".

RC-Have there been changes during the winter months (e.g. retrogressive failure, erosion)?

AC-For the purposes of the present study this information is not relevant. No changes were recorded between the field mapping and the time of the acquisition of the images.

However, after the mapping procedure was completed, a retrogressive movement occurred in the landslide escarpment area. This is visible on the recent images provided by Google Earth.

RC-Is the area cultivated/what is the land cover/land use?

AC-To respond to the question of R1, we modified the text adding the following sentence: "The landslide develops in a crop area, and the lithology consists in a sequence of sand, silt and clay layered deposits."

RC-Add a table specifying what was done by whom and when. Also include the abbreviations of the persons.

AC-We considered carefully the option of adding a table, as suggested by R1. However, we concluded that this was not necessary, and would only add to the length of the paper, without improving clarity or readability. The abbreviation of the individuals who performed the GPS mapping and photointerpretation are given in sections 4.1 and 4.2.

RC-Describe the 2.5D pseudo-stereoscopic data in more detail. Why was the landslide mapping based on the orthorectified UAV-imagery done in Google Earth and not using a more suited GIS software?

AC-We acknowledge that our choice us using Google Earth™ was poorly explained. We have changed and expanded the text, that now reads: "To interpret visually the ultra-resolution UAV image, the interpreter overlaid ("draped") the image on Google Earth™. For the purpose, we first treated the UAV image with the gdal2tiles.py software to obtain a set of image tiles compatible with Google Earth™ terrain visualization platform. To the best of our knowledge, the platform is the only free, 2.5D image visualisation environment that allows the editing of vector (i.e., point, line, polygon) information. Other commercial (e.g., ArcScene) and open source (e.g., ParaView, GRASS GIS), 2.5D visualization tools do not provide editing capabilities. Google Earth™ is a user-friendly solution for mapping single landslides, and for preparing landslide event

inventories for limited areas, with the possibility for the user to visualize a landscape from virtually any viewpoint, facilitating landslide mapping".

RC-Did you use the DEM included in Google Earth for aiding the mapping procedure?

AC-The DEM available in Google Earth™ is low-resolution, pre-event DEM, that does not provide adequate information on the specific landslide morphology. On the other hand, the DEM proves useful to frame the landslide in the general morphology of the slope.

RC-Why didn't you consider a DEM based on the UAV-point cloud?

AC-Indeed, we considered this option carefully. However, to the best of our knowledge, there is no dedicated 2.5D GIS software that allows for editing on a custom DEM used to drape ortho-photographs. The only way to use the DEM based on the UAV-point cloud would have been to use a dedicated GIS for 2.5D visualization software, and a 2D GIS editing environment to transfer the information obtained from the visualization to a base map. The procedure would have introduced an additional source of uncertainty.

RC-Since in most of the scene there is no high vegetation (trees), the landslide's morphology should be represented well. Also other derivatives of the resulting UAVC3 NHESSD Interactive Comment Printer-friendly version Discussion paper based DEM (e.g. shaded reliefs, e.g. Niethammer et al., 2010) could be used for landslide mapping. Then, also the morphometric features could have been mapped better using the UAV data.

AC-The use of maps derived from the elevation data is out of the scope of the work, and of the paper that focuses on optical images. We acknowledge that the scope of the work was not fully clear. When have changed the tithe that now reads "Criteria for the optimal selection of remote sensing optical images to map event landslides". We also added the word "optical" in the Introduction, where we now write: "These maps were compared to an eighth map considered to be the benchmark showing the

"ground truth" i.e., the "true" position, shape and extent of the Assignano landslide. Based on the results of the map comparison, we infer the ability of different optical images, characterized by with different spectral and spatial characteristics, to portray the landslide features that can be exploited for the visual detection and mapping of landslides."

RC-Describe the transfer of mapped landslide features from Google Earth to the GIS. Which GIS software was used?

AC-To transfer the mapped landslide features from Google Earth™ to a GIS database we used the open source GIS software QGIS. The mapping produded in Google Earth™ was imported in QGIS as a Keyhole Markup Language (kml) file, and then converted in the ESRI Shapefile (shp) format.

RC-Which coordinate system/projection was used for the individual datasets (can Google Earth handle ETRF-2000)?

AC-Seven of the dataset were originally mapped in WGS 84 33 N (EPSG 32633). Concerning the question about the capacity of Google Earth to handle ETRF-2000 reference system, we acknowledge that some errors are expected when a raster map is warped on Google Earth, due primarily to the spherical Mercator reference system adopted by Google Earth). However, we did not observe relevant systematic positional errors. This is evident also when comparing the map obtained using the monoscopic UAV image with the map obtained overlaying ("draping") the same image on Google Earth™.

RC-Mention that you mapped the source/transportation area and the deposition area as separate landslide features. How did you discern the source/transportation area from the deposition area?

AC-To respond to this comment of R1, we added language to the paragraph. The new text now reads: "The source and transportation area is bounded locally by subvertical, 2 to 4-m high escarpments. In the landslide, terrain slope averages 11°, and is steeper (12°) in the source and transportation area than in the deposition area (9°). The landslide signature (Pike, 1988) is different in the different parts of the landslide. In the source and transport area the signature is predominantly photographical (radiometric), whereas in the landslide deposit it is mainly morphometric (topographic). The differences allow to separate the source and transportation area from the deposition area".

RC-Are there indicators beyond subjective visual recognition?

AC-We are not sure we understand fully the question. However, we point out that visual recognition is by definition subjective, but it is based on objective and reproducible observations. As stated in section 2, the two landslide portions show different average slope and different photographical and morphological signatures. An expert geomorphologist is able to identify and classify the different landslide signatures, in the source and transport zone and in the deposition area.

RC-How did you treat shadows during landslide mapping?

AC-The images we used were free from shadows. We added language in Section 3 to state that: "Both satellite and UAV images are free from deep shadows (Fig. 2)."

RC-Comment on the comparability of landslide features mapped on different scales (1:1.000 to 1:6.000).

AC-We accepted this comment of R1, and we changed the text adding the following sentence to paragraph 4.2: "The scale of observation was selected to obtain the best readability of each landslide feature and the surroundings, which is a common practice in image visual analysis for landslide mapping (Fiorucci et al., 2011). Hence, even if the maps were produced at slightly different observation scales, the differences arising from the comparison are due to actual features (i.e., the image resolution and radiometry), and not to the different observation scales."

Technical comments AC-We thank R1 for the technical comments. We accepted all the technical comments of R1, and we corrected the text accordingly.

Figures and Tables RC-Figure 1: add information on the shown datasets in Fig. 1A (also add a reference to Google Earth), also specifying the source of the polygons and -lines.

AC-To respond to this request of R1, we added language in the caption, that now reads: "The Assignano landslide, located near Collazzone, Umbria, central Italy. (A) global view of the landslide. (B) detail of the landslide source area. (C) detail of the landslide transportation area. (D) detail of the landslide deposit. Base image obtained overlaying ("draping") the image on Google Earth$^{TM}$. Red line is the boundary of the landslide obtained using the RTK DGPS (benchmark)".

RC-Figure 2: Add a north arrow. Change DGPC to DGPS in the caption. AC-In the new version of the manuscript Figure 2 has become Figure 5. We thank R1 for the suggestion, and we change the figure and the caption accordingly.

RC-Table 1: change meter to metre in the caption

AC-We accepted this suggestion of R1, and amended the caption accordingly.

Reference AC-We added to the list of references the three citations suggested by R1.

Please also note the supplement to this comment:
https://www.nat-hazards-earth-syst-sci-discuss.net/nhess-2017-111/nhess-2017-111-AC1-supplement.pdf

---

## Author Comment (AC2) · 7 Jul 2017

Answer to referee # 2

RC - Referee comments

AC- Author comments

RC-This paper aims at comparing different geological mapping of the perimeter of an Italian landslide within a temperate area partially covered with forested vegetation. The authors realize that high resolution, various wave length and stereoscopic views helps a lot in order to precise the external geometry of some sections of this landslide (crown transport and sedimentation areas). Moreover, authors quantify the misfit in between those different mappings relative to a benchmark (Field RTK DGPS survey) through a

useful error matrix. The differences in the mapping partly derived from the forest cover that hide the exact perimeter of this landslide. To my point of view, the main teaching of this paper is not new as geologists/geomorphologists experts in mapping know since a long time that very high resolution, as well as False color composition (relative to True color) and stereoscopic analyses are major and compulsory keys for a precise and exact geological/geomorphological mapping of any geological/geomorpholoigcal objects. Moreover, planimetric differences of mapped objects also are not new see for instance the work on various fractal distances on the measurements of a Britany shoreline that change a lot function of the scale and the resolution (see the basic work of the mathematician benoit Mandelbrot ENSMP Fontainebleau and his team in the 1980's). The interest of this paper is to illustrate it correctly with a pedagogic example and to recall to any scientists these facts using a specific example. In that sense it is interesting for NHESS to publish it.

AC-We thank this Reviewer (R2) for this comment. R2 is correct in saying that our work (and the paper) article does not introduce novelty concerning the adopted techniques used to recognize and map landslides. Indeed, the purpose of the work is to identify which images characteristics are more suitable to map landslide features. With this respect, and to the best of our knowledge, we maintain that there are not very many examples in the landslide literature. Mapping differences are not related to the presence of vegetation (we are working in a crop area and not in forested terrain), but rather to the ability of the images to highlight the two key landslide features, namely: the morphological and photographical signatures. Moreover, we show that the highest resolution or the FCC may not be the best choice for landslide recognition and mapping. Since landslide features are predominantly morphological, this work shows that it can be preferable to use stereoscopic images with smaller spatial resolution than ultra-resolution monoscopic images.

RC-Could you differentiate more clearly the 3 sections of this landslide on those various mapping erosional part (crown), transport section, and at least the sedimentational

section (toe). With which image (and why) do we have the best and the more exact geological mapping of this landslide?

AC-The best (and "more exact") landslide mapping could be considered the one obtained using stereoscopic satellite TC image for the deposition area (E = 0.21) and the monoscopic UAV image for the source and transportation area (E = 0.15). Overall, and considering the entire landslide, the best mapping (i.e., the one most similar to the benchmark) is the one obtained using Stereoscopic Satellite TC image (E = 0.18). The mentioned numerical values of the error (Error Index proposed by Carrara et al., 1992) are shown in Figure 6. Concerning the choice of a single "best" image, the issue is discussed in the last paragraphs of the "Concluding remarks". The discussion is done from a wider point of view than the investigation of the specific landslide considered in this work. In fact, we conclude that the choice of the best type of image is dictated by technical and cost-related constraints. We stress that this work focuses on the identification of the characteristics of the images that enable the best recognition and mapping of landslide features. Distinguishing between the different kinematic domains of the landslide, or recognizing geological or geotechnical features of the landslide, is out of the scope of this research work.

RC-Could you precise the inputs and differences through local case examples on a new figure of high resolution DTM, FCC and stereoscopic mapping in order that the reader will be able to get an independent position.

AC-To respond to this comment of R2, we added a new figure (Figure 2). In this Figure we show the WorldView-2 images in TC and FCC, and the UAV image. For each image, we also show a detail of the source and of the deposition area. We decided against adding a stereoscopic image, mainly because a printed anaglyph does not provide the same information of a digital stereoscopic system, that is the one used by the geomorphologist to produce the maps. As such, the anaglyph would have provided potentially misleading information. Lastly, we did not use the high-resolution DEM to prepare the landslide maps.

RC-Please finally dealing with your experience on that landslide what (and why) is your best and more exact mapping? please justify it?

AC-We maintain we have already answered to this question of R2.

RC-What is your best methodological solution to map precisely such Italian landslides?

AC-The Assignano landslide represents an instructive, didactic example of a landslide that has both clear photographical and morphological signatures. By using different images, with different spectral and spatial characteristics, and comparing the maps obtained to a defined benchmark, the more accurate and cost-effective mapping is the one obtained by using the UAV image heuristic interpretation method. This is clearly the case if one considers the mapping of just one landslide. We stress that selecting the best mapping of the Assignano landslide is not the goal of this work, as clearly stated in the "Concluding remarks", and specifically in the last paragraph, where we write: "Although we conducted our study on a single landslide (Fig. 1), we maintain that the findings are general, and can be useful to decide on the optimal imagery and technique to be used when planning the production of a landslide inventory map." To further clarify the issue, in the revised version of the manuscript, we added the following sentence in the Introduction: "We maintain that the results obtained in our test case are general, and should be considered for the optimal selection of images for the detection and mapping event landslides.".

RC-If you compare the benchmark and the mappings the map E (stereoscopic image seems the best fit... could you comment on that?

AC-A comparison between the different mappings and the benchmark are shown in Figure 5 and quantified, using the Error Index E, in Figure 6-III. The smallest E value corresponds to Map E. This means that the stereoscopic satellite image with true colours has the characteristics to resolve the photographical and morphological signature of the landslide. Thus, for our test case, it is the best image. When the morphological and photographical features are investigated separately, the best choice is Map

E for the morphological features, and Map G for the photographical features, as shown in Figure 6-II and Figure 6-I, respectively.

RC-Definitely I do not understand the misfit between map A (field DGPS survey) and map B (field landslide mapping), could you comment on the expert's landslide mapping discrepancies?

AC-The field mapping activities consisted in (i) a reconnaissance field survey and (ii) in RTK GPS aided survey are described in detail in Section 4.1. The two mapping methods have inherently different levels of accuracy. The reconnaissance field survey is a multi-step, manual procedure, whereas the RTK GPS aided survey consists in an automatic measurement, with a well-defined accuracy dictated by the D-GPS technology of about 2 to 5 centimeters. The explanation is given in Section 4.1.

into details:

RC-p4. line92-94: precise ...predominantly photogrammetric... and morphometric...

AC-The signature of a landslide is photographical and not photogrammetric. For photographic signature we intend that the landslide is recognizable on the images thanks to photographic characteristic of the image, including tone, colour, tone, mottling, and texture. We change the word "morphometric" with the word "morphological".

RC-p5. line 108: an horizontal...

AC-We do not accept this editorial suggestion of R2. This is because the "h" of "horizontal" is pronounced as an aspirate.

RC-page 6, line 162: field

AC-We thank R2 for this suggestion. We corrected the error accordingly.

RC-page7 line 182: perform an heuristic AC-As before, do not accept this editorial suggestion of R2. This is because the "h" of "horizontal" is pronounced as an aspirate.

RC-page 8, l221: this source area was characterized by small cracks (please show on a figure those features.

To respond to this request of R2, we have added the new Figure 2.

RC-page 9, line 228 to 257 the comments of the table 2 is difficult to follow could you find an easier solution more convenient and easier to understand to present those results?

AC-We maintain that providing (e.g., in Table 2) and describing landslide key and standard measures is useful. For this reason, we have not changed this part of the text.

RC-P.11 , line 282 poor agreement please precise...

AC-We acknowledge the problem, and we chanced the text. In the attempt to clarify the meaning, we now use "high value of the error index" instead of "poor agreement".

RC-P.11, line 287: good agreement please precise...

AC-We acknowledge the problem, and we chanced the text. In the attempt to clarify the meaning, we now use "low value of the error index" instead of "good agreement".

RC-P.13, line 343 please precise a sentence on the resolution of the NIR datasets used herein and what could be the inputs if the NIR dataset if it would have 3x3cm2 ground resolution...

AC-To respond to this comment of R2, we added the following sentence: "We conclude that use of the additional information contributed by the Near Infrared (NIR) band in the 1.84-m resolution satellite image did not improve the quality of the mapping. On the other hand, the contribution of the NIR in the 3-cm UAV image remains unknown."

RC-P.14 line 385 ...is comparable...is to my point of view poor... We do need to have precision on the differences in between mapping from stereoscopic and high resolution... You are working on a local case example you should go farther on your reflection and give to the scientific community your choice of the best way to map such kind of

landslide.

AC-The comparison among the different maps obtained using stereoscopic satellite images and UAV images is supported by the value of the error index E, which is $0.20 \geq E \geq 0.26$ for the entire landslide, $0.21 \geq E \geq 0.29$ for the deposition area, and $0.20 \geq E \geq 0.25$ for the transportation area. The mentioned E values are given in the manuscript, and our conclusions are unambiguously drawn on the basis of the analysis of such values. In particular, the main difference between maps obtained from stereoscopic and UAV images is in the mapping of the deposition area, where the morphological signature of the landslide was better detected using the stereoscopic satellite image than using the ultra-resolution monoscopic images ($0.21 \geq E \geq 0.29$). This is also stated in the "Concluding remarks". We maintain that the selected test case is well representative of the scenarios one may be presented with in the visual mapping of a earthlow. RC-P.14, lines 396-397: and partly independent from the local lighting conditions including the cloud cover... please precise... AC-The acquisition of an UAV image can be planned selecting the best light conditions. This because, most commonly, is the UAV operator that decides when to fly. Also, the flight altitude of a UAV is typically much lower than the clouds height. RC-p.15, l 407 flying AC-We thank R2 for picking up the error. We amended the text accordingly. RC-P.15, l. 412: large or very large areas... AC-To respond to this comment of R2, we modified the text as follow: "Fourth, a comparative analysis of the technological constrains and the costs of acquisition and processing of ultra-resolution imagery taken by UAV, and of high, or very-high resolution imagery taken by optical satellites, revealed that the ultra-resolution images are well suited to map single event landslides, clusters of landslides in a single slope, or a few landslides in nearby slopes in a small area (up to few square kilometres, Giordan et al., 2017) , and prove unsuited to cover large and very large areas where the stereoscopic satellite images provide the most effective option (Boccardo et al., 2015)". RC-p.16, l 447-448: a better resolution and spectral resolution did not contribute significantly to reducing the mapping errors: ??? please precise... AC-R2 is right in saying that the highest resolution images did not provide the best result for the purpose of this work and for

the test case. This is mainly due to the fact that resolution is not the only characteristic of a remotely-sensed image. Other characteristics relevant to landslide recognition and mapping are the stereoscopic view and the spectral content. The outcome of this work shows that stereoscopic view is a key requirement to accurately recognize and map landslide features. In the depositional area, the lowest error is obtained using the stereoscopic satellite images. Even if the UAV images have a spatial resolution higher than the satellite images, the mapping error in the depositional area remains larger than the error obtained using the stereoscopic satellite images. On the other hand, the comparison between the mapping obtained from the stereoscopic satellite images in TC and stereo satellite images in FCC, don't highlight differences, meaning that to map depositional area with mainly morphological signature, stereoscopy is the most important characteristic. To clarify the issue, we added the following sentence: "FCC and TC in the stereoscopic satellite images give similar values of the error. This indicates that the spectral resolution of the images does not provide useful information to recognize and map the landslide morphological features. On the other hand, the high spatial resolution provided by the UAV images reduces the error, when compared to the monoscopic satellite imagery. However, the error obtained using the UAV images remains higher than that obtained using stereoscopic satellite images, despite the latter having a pixel one order of magnitude larger than the UAV images. We conclude that the increase in the spatial resolution improves the ability to map morphological features when using monoscopic images.

RC-p.16, l 461: prove to be very effective

AC-We thank R2 for the suggestion, and we amended the sentence accordingly.

RC-P. 17, 8 l 481 acknowledgments the references needs to be carefully checked.

AC-We checked the acknowledgments the list of references.

RC-Fig.1 to 4: please give comments within the legend that give the key points of the figures.

AC-We changed the captions of Figures 4, 5 are 6 accordingly, to give to the reader a key point of the figure. Figure 4. We add in the caption the following sentence: "The photographs taken in the field and the Google Earth™ image were used to prepare the reconnaissance field map." Figure 5. We added the following sentence to the caption: "Visual inspection of the images reveals the maps most similar to the benchmark." Figure 6. We added the following sentences to the caption: "The error index (E) proposed by Carrara et al. (1992), was used to compare quantitatively the different landslide maps." "E spans the range from 0 (perfect matching) to 1 (complete mismatch)."

RC-Add a figure with specific details inputs of the landslide and compare it to the different geological mappings.

AC-We have added the new Figure 2 to show the WorldView-2 images in TC and FCC, and the UAV image. For each image details of the landslide source and depositional areas are also shown.

Please also note the supplement to this comment:
https://www.nat-hazards-earth-syst-sci-discuss.net/nhess-2017-111/nhess-2017-111-AC2-supplement.pdf

---

## Author Response (AR1)

***Subj.***: Re-submission of manuscript nhess-2017-111

       Dear Paolo Tarolli,

This cover letter is to go with our electronic re-submission of the manuscript *Criteria for the optimal selection of remote sensing images to map event landslides* by Federica Fiorucci, Daniele Giordan, Michele Santangelo, Furio Dutto, Mauro Rossi, Fausto Guzzetti.

We are grateful to you and to the two reviewers for their constructive comments that helped us to improve the work.

In preparing the new version of our work, we considered all the comments and suggestions made by the two referees, which were pertinent and helpful.

To respond to the requests of both the reviewers we modified the Title, and all the other sections according to the reviewer requests. We added three new figures as requested by the reviewers.

We provide a list of our responses to the referee's comments, including details on the changes made to the text.

Overall, we consider this new version of the manuscript significantly improved. We hope the paper can be accepted for publication in the Special Issues: *The use of remotely piloted aircraft systems (RPAS) in monitoring applications and management of natural hazards*.

We look forward to hearing a decision from you soon.

                             Sincerely,
                         Federica Fiorucci, on behalf

*The authors present a study focussing on an expert-based interpretation of imagery data with the aim of mapping landslide features of a single landslide. Various data are tested and the mapping results are compared to reference data and field mappings.*

*The authors then give recommendations regarding the feasibility of the different mapping techniques and imagery data for landslide mapping. The employed methods are standard methods (dGPS, heuristic landslide mapping techniques), so there is no methodical innovation. The used software are commercial products. The results are difficult to reproduce, since only one expert did all the mapping. It would be interesting to see, how the landslide would have been mapped by further experts (>10). Furthermore, it remains unclear if the results are transferable to the relevant scale of event landslide inventories or to other types of landslides.*

We thank this Reviewer (R1) for this comment. As correctly noted by R1, in this work we adopted different (standard) techniques and digital images to produce a landslide inventory. The techniques consist in field mapping and photointerpretation. For the latter we used six different digital products. However, we point out that the aim of the effort was not to investigate the feasibility of techniques, nor to give absolute criteria to choose among different images. The study focuses on the definition of criteria for the selection of remote sensing images for the specific purpose of mapping event landslides. For this reason, we relied upon a single expert to perform the landslide recognition and mapping. We considered the possibility to use more experts. However, this would have added the uncertainty inherent in the subjective interpretation of aerial photography for landslide mapping (see e.g., Carrara et al., 1992, Uncertainty in evaluating landslide hazard and risk. ITC Journal, 172–183). The uncertainty inherent with the interpreters would have mixed (and covered partially) the "signal" from the different imagery used for our experiment. Since the scope of the research was to investigate the information content of the imagery (and not of the interpreters) we ruled out the possibility of using more interpreters. Further, the researcher geomorphologist who interpreted the images and prepared the maps (MS) has a significant experience in photointerpretation for landslide mapping (he has prepared 25 landslides maps, including event maps, geomorphological maps, multi-temporal maps, covering more than 4000 km$^2$, obtained using both monoscopic and stereoscopic satellite images and stereoscopic aerial photographs). Thanks to the expertise of the mapper, in each digital image the relevant features of the landslide were recognized fully. Thus, we are confident that differences among the six maps are to be ascribed to the sole resolving power of the different images. We have clarified this point in the text (see below). Moreover, we selected a landslide having both morphological and photographical signatures, which are the two key features that allows to recognize and map landslides from digital images. For this reason, we maintain that the results we have obtained are valid at all scales, and for most landslide types.

*The text is generally well written, but there are some minor mistakes of grammar and style. Some of them are addressed below, but it would be out of scope to raise every issue. Therefore, I recommend careful copy editing. Below, I focus on issues concerning the scientific content of the manuscript. Where numbers are given in the specific comments*

*they refer to the manuscript page and line. A major revision carefully addressing the raised issues below is required, before the paper can be considered for publication.*

We thank R1 for reading carefully or Manuscript. We amended the text following R1 suggestions, where applicable.

***Specific Comments***

*Consider introducing the principle of heuristic, visual mapping of landslide features based on the interpretation of landslide signs ('geometric signature'; Pike, 1988). This is well explained in Section 4.2. However, an introductory description of the procedure would benefit the understanding of the reader. In this context, also explain the advantage of stereoscopic over monoscopic interpretation techniques.*

In the Introduction, we added the following language to clarify the text:

"The heuristic visual mapping of landslide features is based on the systematic analysis of image photographic and morphological characteristics such as colour, tone, mottling, texture, shape, size, curvature (Pike 1988). These photographic and morphological characteristics encompasses all the possible landslide features that can be used for the (visual) interpretation of the available imagery."

*Consider addressing the necessary positional accuracy of the mapped landslide features with respect to the intended use/scale of the compiled landslide inventory.*

We accepted this suggestion of R1. In the Discussion section, we added the following sentence:

"Where possible, we recommend that the acquisition of images used for the production of event, seasonal or multi-temporal landslide inventory maps is planned considering the typical landslide signature, in addition to the purpose (event inventory, planning of monitoring systems), scale of the mapping (i.e. regional or slope scale), and the size and complexity of the study area (see Table 3)."

*Consider adding a sentence addressing the potential of UAV-based imagery for efficiently analysing changes over time (e.g. Turner et al., 2015). Added to Discussion section.*

We accepted this suggestion of R1. In the Discussion, we added the following sentence:

"The use of UAV images was recently proposed by Turner et al. (2015) for determining the landslide dynamics, exploiting time series of images that can be constructed using UAVs. The result is achievable thanks to centimetre co-registration accuracy of the UAV images."

*Which type of landslide is it, what type of material is involved (since the area seems to be not very steep) and what are the causes and failure mechanism?*

We accepted the comment of R1, and changed the text as follows:

"For our study, we selected the Assignano landslide, a slide-earthflow (Hutchinson, 1970) triggered by intense rainfall in December 2013 in the northwest-facing slope of the Assignano village, Umbria, central Italy (Fig. 1). The landslide develops in a crop area, where a layered sequence of sand, silt and clay deposits crop out (Santangelo et al., 2015)".

*Have there been changes during the winter months (e.g. retrogressive failure, erosion)?*

For the purposes of the present study this information is not relevant. No changes were recorded between the field mapping and the time of the acquisition of the images. However, after the mapping procedure was completed, a retrogressive movement occurred in the landslide escarpment area. This is visible on the recent images provided by Google Earth.

*Is the area cultivated/what is the land cover/land use?*

To respond to the question of R1, we modified the text adding the following sentence:

"The landslide develops in a crop area, and the lithology consists in a sequence of sand, silt and clay layered deposits."

*Add a table specifying what was done by whom and when. Also include the abbreviations of the persons.*

We considered carefully the option of adding a table, as suggested by R1. However, we concluded that this was not necessary, and would only add to the length of the paper, without improving clarity or readability. The abbreviation of the individuals who performed the GPS mapping and photointerpretation are given in sections 4.1 and 4.2.

*Describe the 2.5D pseudo-stereoscopic data in more detail. Why was the landslide mapping based on the orthorectified UAV-imagery done in Google Earth and not using a more suited GIS software?*

We acknowledge that our choice us using Google Earth™ was poorly explained. We have changed and expanded the text, that now reads:

"To interpret visually the ultra-resolution UAV image, the interpreter overlaid ("draped") the image on Google Earth™. For the purpose, we first treated the UAV image with the gdal2tiles.py software to obtain a set of image tiles compatible with Google Earth™ terrain visualization platform. To the best of our knowledge, the platform is the only free, 2.5D image visualisation environment that allows the editing of vector (i.e., point, line, polygon) information. Other commercial (e.g., ArcScene) and open source (e.g., ParaView, GRASS GIS), 2.5D visualization tools do not provide editing capabilities. Google Earth™ is a user-friendly solution for mapping single landslides, and for preparing landslide event inventories for limited areas, with the possibility for the user to visualize a landscape from virtually any viewpoint, facilitating landslide mapping".

*Did you use the DEM included in Google Earth for aiding the mapping procedure?*

The DEM available in Google Earth™ is low-resolution, pre-event DEM, that does not provide adequate information on the specific landslide morphology. On the other hand, the DEM proves useful to frame the landslide in the general morphology of the slope.

*Why didn't you consider a DEM based on the UAV-point cloud?*

Indeed, we considered this option carefully. However, to the best of our knowledge, there is no dedicated 2.5D GIS software that allows for editing on a custom DEM used to drape ortho-photographs. The only way to use the DEM based on the UAV-point cloud would have been to use a dedicated GIS for 2.5D visualization software, and a 2D GIS editing environment to transfer the information obtained from the visualization to a base map. The procedure would have introduced an additional source of uncertainty.

*Since in most of the scene there is no high vegetation (trees), the landslide's morphology should be represented well. Also other derivatives of the resulting UAVC3 NHESSD Interactive Comment Printer-friendly version Discussion paper based DEM (e.g. shaded reliefs, e.g. Niethammer et al., 2010) could be used for landslide mapping. Then, also the morphometric features could have been mapped better using the UAV data.*

The use of maps derived from the elevation data is out of the scope of the work, and of the paper that focuses on optical images. We acknowledge that the scope of the work was not fully clear. When have changed the tithe that now reads "Criteria for the optimal selection of remote sensing optical images to map event landslides". We also added the word "optical" in the Introduction, where we now write:

"These maps were compared to an eighth map considered to be the benchmark showing the "ground truth" i.e., the "true" position, shape and extent of the Assignano landslide. Based on the results of the map comparison, we infer the ability of different optical images, characterized by with different spectral and spatial characteristics, to portray the landslide features that can be exploited for the visual detection and mapping of landslides."

*Describe the transfer of mapped landslide features from Google Earth to the GIS. Which GIS software was used?*

To transfer the mapped landslide features from Google Earth™ to a GIS database we used the open source GIS software QGIS. The mapping produded in Google Earth™ was imported in QGIS as a Keyhole Markup Language (kml) file, and then converted in the ESRI Shapefile (shp) format.

*Which coordinate system/projection was used for the individual datasets (can Google Earth handle ETRF-2000)?*

Seven of the dataset were originally mapped in WGS 84 33 N (EPSG 32633). Concerning the question about the capacity of Google Earth to handle ETRF-2000 reference system, we acknowledge that some errors are expected when a raster map is warped on Google Earth, due primarily to the spherical Mercator reference system adopted by Google Earth). However, we did not observe relevant systematic positional errors. This is evident also when comparing the map obtained using the monoscopic UAV image with the map obtained overlaying ("draping") the same image on Google Earth™.

*Mention that you mapped the source/transportation area and the deposition area as separate landslide features. How did you discern the source/transportation area from the deposition area?*

To respond to this comment of R1, we added language to the paragraph. The new text now reads:

"The source and transportation area is bounded locally by sub-vertical, 2 to 4-m high escarpments. In the landslide, terrain slope averages 11°, and is steeper (12°) in the source and transportation area than in the deposition area (9°). The landslide signature (Pike, 1988) is different in the different parts of the landslide. In the source and transport area the signature is predominantly photographical (radiometric), whereas in the landslide deposit it is mainly morphometric (topographic). The differences allow to separate the source and transportation area from the deposition area".

*Are there indicators beyond subjective visual recognition?*

We are not sure we understand fully the question. However, we point out that visual recognition is by definition subjective, but it is based on objective and reproducible observations. As stated in section 2, the two landslide portions show different average slope and different photographical and morphological signatures. An expert geomorphologist is able to identify and classify the different landslide signatures, in the source and transport zone and in the deposition area.

*How did you treat shadows during landslide mapping?*

The images we used were free from shadows.
We added language in Section 3 to state that:

"Both satellite and UAV images are free from deep shadows (**Fig. 2**)."

*Comment on the comparability of landslide features mapped on different scales (1:1.000 to 1:6.000).*

We accepted this comment of R1, and we changed the text adding the following sentence to paragraph 4.2:

"The scale of observation was selected to obtain the best readability of each landslide feature and the surroundings, which is a common practice in image visual analysis for landslide mapping (Fiorucci et al., 2011). Hence, even if the maps were produced at slightly different observation scales, the differences arising from the comparison are due to actual features (i.e., the image resolution and radiometry), and not to the different observation scales."

***Technical comments***

We thank R1 for the technical comments. We accepted all the technical comments of R1, and we corrected the text accordingly.

***Figures and Tables***

*Figure 1: add information on the shown datasets in Fig. 1A (also add a reference to Google Earth), also specifying the source of the polygons and -lines.*

To respond to this request of R1, we added language in the caption, that now reads:

"The Assignano landslide, located near Collazzone, Umbria, central Italy. (A) global view of the landslide. (B) detail of the landslide source area. (C) detail of the landslide transportation area. (D) detail of the landslide deposit. Base image obtained overlaying ("draping") the image on Google Earth™. Red line is the boundary of the landslide obtained using the RTK DGPS (benchmark)".

*Figure 2: Add a north arrow. Change DGPC to DGPS in the caption.*

In the new version of the manuscript Figure 2 has become Figure 5. We thank R1 for the suggestion, and we change the figure and the caption accordingly.

*Table 1: change meter to metre in the caption*

We accepted this suggestion of R1, and amended the caption accordingly.

***Reference***

We added to the list of references the three citations suggested by R1.

**Answer to referee # 2**

*This paper aims at comparing different geological mapping of the perimeter of an Italian landslide within a temperate area partially covered with forested vegetation. The authors realize that high resolution, various wave length and stereoscopic views helps a lot in order to precise the external geometry of some sections of this landslide (crown transport and sedimentation areas). Moreover, authors quantify the misfit in between those different mappings relative to a benchmark (Field RTK DGPS survey) through a useful error matrix. The differences in the mapping partly derived from the forest cover that hide the exact perimeter of this landslide.*

*To my point of view, the main teaching of this paper is not new as geologists/geomorphologists experts in mapping know since a long time that very high resolution, as well as False color composition (relative to True color) and stereoscopic analyses are major and compulsory keys for a precise and exact geological/geomorphological mapping of any geological/geomorpholoigcal objects. Moreover, planimetric differences of mapped objects also are not new see for instance the work on various fractal distances on the measurements of a Britany shoreline that change a lot function of the scale and the resolution (see the basic work of the mathematician benoit Mandelbrot ENSMP Fontainebleau and his team in the 1980's). The interest of this paper is to illustrate it correctly with a pedagogic example and to recall to any scientists these facts using a specific example. In that sense it is interesting for NHESS to publish it.*

We thank this Reviewer (R2) for this comment. R2 is correct in saying that our work (and the paper) article does not introduce novelty concerning the adopted techniques used to recognize and map landslides. Indeed, the purpose of the work is to identify which images characteristics are more suitable to map landslide features. With this respect, and to the best of our knowledge, we maintain that there are not very many examples in the landslide literature. Mapping differences are not related to the presence of vegetation (we are working in a crop area and not in forested terrain), but rather to the ability of the images to highlight the two key landslide features, namely: the morphological and photographical signatures. Moreover, we show that the highest resolution or the FCC may not be the best choice for landslide recognition and mapping. Since landslide features are predominantly morphological, this work shows that it can be preferable to use stereoscopic images with smaller spatial resolution than ultra-resolution monoscopic images.

*Could you differentiate more clearly the 3 sections of this landslide on those various mapping erosional part (crown), transport section, and at least the sedimentational section (toe). With which image (and why) do we have the best and the more exact geological mapping of this landslide?*

The best (and "more exact") landslide mapping could be considered the one obtained using stereoscopic satellite TC image for the deposition area (E = 0.21) and the monoscopic UAV image for the source and transportation area (E = 0.15). Overall, and considering the entire landslide, the best mapping (i.e., the one most similar to the benchmark) is the one obtained using Stereoscopic Satellite TC image (E = 0.18). The mentioned numerical values of the error (Error Index proposed by Carrara et al., 1992)

are shown in Figure 6. Concerning the choice of a single "best" image, the issue is discussed in the last paragraphs of the "Concluding remarks". The discussion is done from a wider point of view than the investigation of the specific landslide considered in this work. In fact, we conclude that the choice of the best type of image is dictated by technical and cost-related constraints. We stress that this work focuses on the identification of the characteristics of the images that enable the best recognition and mapping of landslide features. Distinguishing between the different kinematic domains of the landslide, or recognizing geological or geotechnical features of the landslide, is out of the scope of this research wor

*Could you precise the inputs and differences through local case examples on a new figure of high resolution DTM, FCC and stereoscopic mapping in order that the reader will be able to get an independent position.*

To respond to this comment of R2, we added a new figure (Figure 2). In this Figure we show the WorldView-2 images in TC and FCC, and the UAV image. For each image, we also show a detail of the source and of the deposition area. We decided against adding a stereoscopic image, mainly because a printed anaglyph does not provide the same information of a digital stereoscopic system, that is the one used by the geomorphologist to produce the maps. As such, the anaglyph would have provided potentially misleading information. Lastly, we did not use the high-resolution DEM to prepare the landslide maps.

*Please finally dealing with your experience on that landslide what (and why) is your best and more exact mapping? please justify it?*

We maintain we have already answered to this question of R2.

*What is your best methodological solution to map precisely such Italian landslides?*

The Assignano landslide represents an instructive, didactic example of a landslide that has both clear photographical and morphological signatures. By using different images, with different spectral and spatial characteristics, and comparing the maps obtained to a defined benchmark, the more accurate and cost-effective mapping is the one obtained by using the UAV image heuristic interpretation method. This is clearly the case if one considers the mapping of just one landslide. We stress that selecting the best mapping of the Assignano landslide is not the goal of this work, as clearly stated in the "Concluding remarks", and specifically in the last paragraph, where we write:

 "Although we conducted our study on a single landslide (Fig. 1), we maintain that the findings are general, and can be useful to decide on the optimal imagery and technique to be used when planning the production of a landslide inventory map."

To further clarify the issue, in the revised version of the manuscript, we added the following sentence in the Introduction:

 "We maintain that the results obtained in our test case are general, and should be considered for the optimal selection of images for the detection and mapping event landslides.".

*If you compare the benchmark and the mappings the map E (stereoscopic image seems the best fit... could you comment on that?*

A comparison between the different mappings and the benchmark are shown in Figure 5 and quantified, using the Error Index $E$, in Figure 6-III. The smallest $E$ value corresponds to Map E. This means that the stereoscopic satellite image with true colours has the characteristics to resolve the photographical and morphological signature of the landslide. Thus, for our test case, it is the best image. When the morphological and photographical features are investigated separately, the best choice is Map E for the morphological features, and Map G for the photographical features, as shown in Figure 6-II and Figure 6-I, respectively.

*Definitely I do not understand the misfit between map A (field DGPS survey) and map B (field landslide mapping), could you comment on the expert's landslide mapping discrepancies?*

The field mapping activities consisted in (i) a reconnaissance field survey and (ii) in RTK GPS aided survey are described in detail in Section 4.1. The two mapping methods have inherently different levels of accuracy. The reconnaissance field survey is a multi-step, manual procedure, whereas the RTK GPS aided survey consists in an automatic measurement, with a well-defined accuracy dictated by the D-GPS technology of about 2 to 5 centimeters. The explanation is given in Section 4.1.

**into details:**

*p4. line92-94: precise ...predominantly photogrammetric... and morphometric...*

The signature of a landslide is photographical and not photogrammetric. For photographic signature we intend that the landslide is recognizable on the images thanks to photographic characteristic of the image, including tone, colour, tone, mottling, and texture. We change the word "morphometric" with the word "morphological".

*p5. line 108: an horizontal...*

We do not accept this editorial suggestion of R2. This is because the "h" of "horizontal" is pronounced as an aspirate.

*page 6, line 162: field*

We thank R2 for this suggestion. We corrected the error accordingly.

*page7 line 182: perform an heuristic*

As before, do not accept this editorial suggestion of R2. This is because the "h" of "horizontal" is pronounced as an aspirate.

*page 8, l221: this source area was characterized by small cracks (please show on a figure those features.*

To respond to this request of R2, we have added the new Figure 2.

*page 9, line 228 to 257 the comments of the table 2 is difficult to follow could you find an easier solution more convenient and easier to understand to present those results?*

We maintain that providing (e.g., in Table 2) and describing landslide key and standard measures is useful. For this reason, we have not changed this part of the text.

*P.11 , line 282 poor agreement please precise...*

We acknowledge the problem, and we chanced the text. In the attempt to clarify the meaning, we now use "high value of the error index" instead of "poor agreement".

*P.11, line 287: good agreement please precise...*

We acknowledge the problem, and we chanced the text. In the attempt to clarify the meaning, we now use "low value of the error index" instead of "good agreement".

*P.13, line 343 please precise a sentence on the resolution of the NIR datasets used herein and what could be the inputs if the NIR dataset if it would have 3x3cm$^2$ ground resolution...*

To respond to this comment of R2, we added the following sentence:

"We conclude that use of the additional information contributed by the Near Infrared (NIR) band in the 1.84-m resolution satellite image did not improve the quality of the mapping. On the other hand, the contribution of the NIR in the 3-cm UAV image remains unknown."

*P.14 line 385 ...is comparable...is to my point of view poor... We do need to have precision on the differences in between mapping from stereoscopic and high resolution... You are working on a local case example you should go farther on your reflection and give to the scientific community your choice of the best way to map such kind of landslide.*

The comparison among the different maps obtained using stereoscopic satellite images and UAV images is supported by the value of the error index $E$, which is $0.20 \geq E \geq 0.26$ for the entire landslide, $0.21 \geq E \geq 0.29$ for the deposition area, and $0.20 \geq E \geq 0.25$ for the transportation area. The mentioned $E$ values are given in the manuscript, and our conclusions are unambiguously drawn on the basis of the analysis of such values. In particular, the main difference between maps obtained from stereoscopic and UAV images is in the mapping of the deposition area, where the morphological signature of the landslide was better detected using the stereoscopic satellite image than using the ultra-resolution monoscopic images ($0.21 \geq E \geq 0.29$). This is also stated in the "Concluding remarks". We maintain that the selected test case is well representative of the scenarios one may be presented with in the visual mapping of a earthlow.

*P.14, lines 396-397: and partly independent from the local lighting conditions including the cloud cover... please precise...*

The acquisition of an UAV image can be planned selecting the best light conditions. This because, most commonly, is the UAV operator that decides when to fly. Also, the flight altitude of a UAV is typically much lower than the clouds height.

*p.15, l 407 flying*

We thank R2 for picking up the error. We amended the text accordingly.

*P.15, l. 412: large or very large areas...*

To respond to this comment of R2, we modified the text as follow:

"Fourth, a comparative analysis of the technological constrains and the costs of acquisition and processing of ultra-resolution imagery taken by UAV, and of high, or very-high resolution imagery taken by optical satellites, revealed that the ultra-resolution images are well suited to map single event landslides, clusters of landslides in a single slope, or a few landslides in nearby slopes in a small area (up to few square kilometres, Giordan et al., 2017) , and prove unsuited to cover large and very large areas where the stereoscopic satellite images provide the most effective option (Boccardo et al., 2015)".

*p.16, l 447-448: a better resolution and spectral resolution did not contribute significantly to reducing the mapping errors: ??? please precise...*

R2 is right in saying that the highest resolution images did not provide the best result for the purpose of this work and for the test case. This is mainly due to the fact that resolution is not the only characteristic of a remotely-sensed image. Other characteristics relevant to landslide recognition and mapping are the stereoscopic view and the spectral content. The outcome of this work shows that stereoscopic view is a key requirement to accurately recognize and map landslide features. In the depositional area, the lowest error is obtained using the stereoscopic satellite images. Even if the UAV images have a spatial resolution higher than the satellite images, the mapping error in the depositional area remains larger than the error obtained using the stereoscopic satellite images. On the other hand, the comparison between the mapping obtained from the stereoscopic satellite images in TC and stereo satellite images in FCC, don't highlight differences, meaning that to map depositional area with mainly morphological signature, stereoscopy is the most important characteristic. To clarify the issue, we added the following sentence:

"FCC and TC in the stereoscopic satellite images give similar values of the error. This indicates that the spectral resolution of the images does not provide useful information to recognize and map the landslide morphological features. On the other hand, the high spatial resolution provided by the UAV images reduces the error, when compared to the monoscopic satellite imagery. However, the error obtained using the UAV images remains higher than that obtained using stereoscopic satellite images, despite the latter having a pixel one order of magnitude larger than the UAV images. We conclude that the increase in the spatial resolution improves the ability to map morphological features when using monoscopic images.

p.16, l 461: *prove to be very effective*

We thank R2 for the suggestion, and we amended the sentence accordingly.

*P. 17, 8 l 481 acknowledgments the references needs to be carefully checked.*

We checked the acknowledgments the list of references.

*Fig.1 to 4: please give comments within the legend that give the key points of the figures.*

We changed the captions of Figures 4, 5 are 6 accordingly, to give to the reader a key point of the figure.

Figure 4. We add in the caption the following sentence:

"The photographs taken in the field and the Google Earth™ image were used to prepare the reconnaissance field map."

Figure 5. We added the following sentence to the caption:

"Visual inspection of the images reveals the maps most similar to the benchmark."

Figure 6. We added the following sentences to the caption:

"The error index ($E$) proposed by Carrara et al. (1992), was used to compare quantitatively the different landslide maps."

*"E* spans the range from 0 (perfect matching) to 1 (complete mismatch)."

*Add a figure with specific details inputs of the landslide and compare it to the different geological mappings.*

We have added the new Figure 2 to show the WorldView-2 images in TC and FCC, and the UAV image. For each image details of the landslide source and depositional areas are also shown.

[revised manuscript text omitted]

**Figura**

**Figure 3**

**I — Source and transportation area**

Error scale: 0 (green) to 1 (red)

Field mapping: RTK (Map A), Vis (Map B)
Image visual interpretation: WolrdView 2D — TC (Map C), FCC (Map D); WolrdView 3D — TC (Map E), FCC (Map F); UAV 2D — TC (Map G); UAV 2.5D — TC (Map H)

| | Map A | Map B | Map C | Map D | Map E | Map F | Map G | Map H |
|---|---|---|---|---|---|---|---|---|
| Map A (RTK) | | 0.45 | 0.38 | 0.30 | 0.23 | 0.21 | 0.15 | 0.16 |
| Map B (VIS) | | | 0.63 | 0.58 | 0.45 | 0.43 | 0.51 | 0.45 |
| Map C (TC) | | | | 0.17 | 0.44 | 0.43 | 0.38 | 0.43 |
| Map D (FCC) | | | | | 0.38 | 0.36 | 0.31 | 0.37 |
| Map E (TC) | | | | | | 0.14 | 0.25 | 0.20 |
| Map F (FCC) | | | | | | | 0.26 | 0.20 |
| Map G (TC) | | | | | | | | 0.15 |
| Map H (TC) | | | | | | | | |

**II — Deposition area**

| | Map A | Map B | Map C | Map D | Map E | Map F | Map G | Map H |
|---|---|---|---|---|---|---|---|---|
| Map A (RTK) | | 0.67 | 0.52 | 0.50 | 0.21 | 0.27 | 0.28 | 0.30 |
| Map B (VIS) | | | 0.44 | 0.63 | 0.69 | 0.69 | 0.73 | 0.72 |
| Map C (TC) | | | | 0.33 | 0.59 | 0.63 | 0.54 | 0.58 |
| Map D (FCC) | | | | | 0.55 | 0.60 | 0.43 | 0.49 |
| Map E (TC) | | | | | | 0.18 | 0.29 | 0.21 |
| Map F (FCC) | | | | | | | 0.38 | 0.31 |
| Map G (TC) | | | | | | | | 0.15 |
| Map H (TC) | | | | | | | | |

**III — Entire landslide**

| | Map A | Map B | Map C | Map D | Map E | Map F | Map G | Map H |
|---|---|---|---|---|---|---|---|---|
| Map A (RTK) | | 0.55 | 0.45 | 0.40 | 0.18 | 0.20 | 0.19 | 0.19 |
| Map B (VIS) | | | 0.45 | 0.57 | 0.57 | 0.57 | 0.59 | 0.59 |
| Map C (TC) | | | | 0.29 | 0.48 | 0.50 | 0.46 | 0.47 |
| Map D (FCC) | | | | | 0.42 | 0.44 | 0.35 | 0.37 |
| Map E (TC) | | | | | | 0.15 | 0.21 | 0.20 |
| Map F (FCC) | | | | | | | 0.26 | 0.25 |
| Map G (TC) | | | | | | | | 0.08 |
| Map H (TC) | | | | | | | | |

[Figure]

| A-B | ΔX (m) | ΔY (m) |
|---|---|---|
| ◇ | -1.1 | 1.1 |
| ○ | 0.3 | 1.3 |
| △ | -0.4 | 1.2 |
| ✳ | 1.7 | -1.8 |
| ✖ | 2.1 | 2.9 |
| ▱ | 0.3 | 1.3 |
| ✦ | -0.8 | -2.4 |

**Figure 4**

[Figure]

743 **Figure 5**

[Figure]

**Figure 6**

_______________________

_______________________

**Figure 7**

[Figure]

---

## Author Response (AR2)

***Subj.***: Re-submission of manuscript nhess-2017-111

Dear Paolo Tarolli,

This cover letter is to go with our electronic re-submission of the manuscript *Criteria for the optimal selection of remote sensing images to map event landslides* by Federica Fiorucci, Daniele Giordan, Michele Santangelo, Furio Dutto, Mauro Rossi, Fausto Guzzetti.

We are grateful to you and to the reviewer for their constructive comments that helped us to improve the work.

In preparing the new version of our work, we considered all the comments and suggestions made by the referee, which were pertinent and helpful.

To respond to the requests of both the reviewers we modified the Abstract, and all the other sections according to the reviewer requests.

We provide a list of our responses to the referee's comments, including details on the changes made to the text.

Overall, we consider this new version of the manuscript significantly improved. We hope the paper can be accepted for publication in the Special Issue: *The use of remotely piloted aircraft systems (RPAS) in monitoring applications and management of natural hazards*.

We look forward to hearing a decision from you soon.

Sincerely,
Federica Fiorucci, on behalf

**Answares to Reviwer 3**

*The paper is an interesting contribution to the journal. However, it should be pointed out more carefully that it is a showcase of a technical application.*

AC- We thank the reviewer for pointing out the value of the contribution to the journal. However, we disagree in considering it as a showcase of a technical application, since the experiment was designed to evaluate the impact of images characteristics on landslide mapping. For how the experiment was conceived and developed, the findings are general and applicable to all the cases similar to the setting of the experiment.

*Overall, I only have few major points to highlight and some minor issues as they arose during the reading of the research.*

*1.The title speaks about criteria for the selection of images. However, the criteria are not well defined in the manuscript, and they are only slightly mentioned at the very end of the conclusions. If the point of the manuscript is indeed to present some criteria for selection, these criteria should appear more clearly (e.g. in a list? A well define paragraph?)*

AC- In the paper, the criteria are well explained in **Table 3**, and commented in the Discussion section and resumed in the "Concluding remarks" section. The reason why they appear in the "Discussion" section, is that the criteria are a consequence of the results of the experiment. The text in the Discussion section extensively comments on the advantages and limitation of using the different types of images taken into account in the experiment for landslide mapping purposes. The section reads:

> "For event landslide mapping, selection between ultra-resolution pseudo-stereoscopic UAV images and very-high resolution stereoscopic satellite images depends on (i) the extent of the investigated area, (ii) the available resources, including time and budget, and (iii) the accessibility to the study area. The selection is largely independent from the landslide signature, at least for landslides similar to the Assignano landslide. From an operational perspective, modern multi-rotor UAVs allow for the acquisition of ultra-resolution images over small areas in a limited time, and at very low costs. UAV-based surveys are flexible in their acquisition planning, and partly independent from the local lighting conditions, including the cloud cover. As a drawback, UAVs are strongly (and negatively) affected by wind speed and weather conditions, they allow for a limited flight time (currently approximately 20 minutes in optimal conditions), which is reduced in bad weather conditions and in cold environments, and typically have limited data storage capacity. Further, it must be possible for the pilot to be at the same time near to the area to be surveyed and to maintain a safe distance from the UAV, a condition that may be difficult to attain in remote or in mountain areas. Collectively, the intrinsic advantages and limitations of modern UAVs make the technology potentially well suited for the acquisition of ultra-resolution images for event, seasonal, and multi-temporal mapping of single landslides, of multiple landslides in a single slope, or in a relatively small area (a few hectares). The use of UAV images was recently proposed by Turner et al. (2015) for determining the landslide dynamics, exploiting time series of images that can be constructed using UAVs. The result is achievable thanks to centimetre co-registration accuracy of the UAV images. Use of UAVs becomes impracticable with the increasing extent of the study area, largely due to (i) the operational difficulty of flying UAVs over large areas (more than a few square kilometres), and (ii) the acquisition and image processing time and associated cost, which increase rapidly with the size of the study area (Table 3). On the other hand, very-high resolution, stereoscopic satellite images have also advantages and limitations for the production of event, seasonal and multi-temporal landslide inventory maps (Guzzetti et al., 2012). The main advantage of the satellite images is that they cover large or very large areas (tens to hundreds of square kilometres) in a single frame with a sub-metre resolution well suited for landslide mapping through the expert visual interpretation of the images (Ardizzone et al., 2013). On the other hand, limitations remain due to distortions caused by different off-nadir angles in successive scenes, and to difficulties – in places severe – to obtaining suitable (e.g., cloud-free) images at the required time intervals. This is particularly problematic for the production of seasonal and multi-temporal landslide maps. Information on the photographic or morphological signature of the typical, or most abundant, landslides in an area, is important to selecting the optimal characteristics of the images best suited for the production of an event, seasonal or multi-temporal landslide inventory map. Use of images of non-optimal characteristics for a typical landslide signature in an area may condition the quality (completeness, positional and thematic accuracy) of the landslide inventory. Where possible, we recommend that the acquisition of images used for the production of event, seasonal or multi-temporal landslide inventory maps is planned considering the typical landslide signature, in addition to the purpose (event inventory, planning of monitoring systems), scale of the mapping (regional or slope scale), and the size and complexity of the study area (Table 3)."

Moreover, for more clarity we added a text at the end of the "Introduction "section that reads:

"Arguably, the combination of images characteristics, the prevalent landslide signature, the size of the study area, and the available resources define the criteria for the optimal selection of remote sensing images for landslide mapping."

*2. English needs polishing and revision. Many times the authors overuse 'i.e.' or they use sentences in a 'personal' approach ("we did this", "we highlight that", rather than describing what the research indicates), and many sentences are very long and hard to follow.*

AC- We thank this Reviewer (R3) for this comment. Most of the 'i.e' were removed, and most of the sentences with a personal approach were converted in an impersonal form. We also revised the text, removing and simplifying the very long sentences.

*3. The work does not compare eight maps. It compares seven maps. The dGPS survey is the benchmark (or ground truth), so it is misleading to include it in the list.*

AC-Thank you for the suggestion.

The text was modified accordingly.

*4. One of the main findings of this work is that a photographic landslide signature is best mapped with higher resolution images, while morphometric signatures are better identified with stereoscopic images. However, the reader does not know what photographic signatures VS morphometric ones are (this is never described in the manuscript).*

AC- We thank the Reviewer for this comment. We described more clearly what morphological and photographical signature are in the following sentences (which give also credits of other works in the literature):

In the "Introduction" section we state:

"Heuristic visual mapping of landslide features is based on the systematic analysis of photographic characteristics such as colour, tone, mottling, texture, shape, and morphological characteristics such as size, curvature, concavity and convexity (Pike, 1988). The mentioned photographic and morphological characteristics encompass all the possible landslide features that can be used for the (visual) image interpretation.."

In the "Study area" section, we state:

"The landslide signature (Pike, 1988) is different in the different parts of the landslide. In the source and transportation area the signature is predominantly photographical (radiometric), whereas in the landslide deposit it is mainly morphological (topographic). The photographic signature consists in all the landslide features that can be detected by the analysis of the photographical characteristics of a given image: colour, tone, pattern and mottling of a given image (Guzzetti et al., 2012). The morphological signature consists in all the landslide features that can be detected by the analysis of the topography, therefore, features such as curvatures, shape, slope, concavity and convexity are always taken into account (Guzzetti et al., 2012). The differences within the landslide allowed to separate the source and transportation area from the deposition area."

*5. The discussion chapter can be shortened and reorganised; much of it is a repetition of the previous chapter, while the reader would expect to find here some additional considerations about the meaning of the results.*

AC- As suggested by the reviewer, we shortened the discussion section, removing most of the first paragraph. Moreover, to increase the readability of the paragraph, all the values between brackets related to the Error value (a repetition of the results) were removed. Admittedly, we disagree for what concerns the considerations on the meaning of the results, which are quite extensively described in the text.

*Minor comments*

*Abstract*

*The abstract needs rewording. It should be more research-oriented, and less of a description of the team effort. E.g. first sentence 'we executed….' Could be rephrased to 'this paper presents…'*

*The work, furthermore, does not compare eight maps, but rather seven different maps as compared to a reference dGPS survey. The so called '8th map', being the ground-truth, is not part of the considered dataset, but it is the benchmark used to compare all the others.*

AC-Thanks to R3 for the suggestion. We reorganized the abstract removing most of the personal form sentences.

*Lines 19 to 24 report a very long sentence, that needs rephrasing. E.g. "Six maps were obtained through expert knowledge by visual interpretation of images from different sources taken on April 14, 2014. The dataset comprised monoscopic and pseudo-stereoscopic (2.5D) ultra-resolution (0.3 × 0.3 m) images derived using a Canon EOS M photographic camera mounted on a CarbonCore 950 hexacopter, and monoscopic and stereoscopic true-colour and false-colour- composite images (1.84 × 1.84 m resolution) taken by the WorldView-2 satellite."*

AC- In the present version of the abstract this sentence has been removed.

*Introduction*

*Line 48 > 'to support' should be changed to 'supporting the installation of…'*

AC-We thank R3 for this suggestion. We corrected the text accordingly.

*Line 52 to 58 > "…through field surveys (Santangelo et al., 2010) or the heuristic visual interpretation of monoscopic or stereoscopic aerial or satellite images (Brardinoni et al., 2003; Fiorucci et al., 2011; Ardizzone et al., 2013), of LiDAR-derived images (Ardizzone et al., 2007; Van Den Eeckhaut et al., 2007; Haneberg et al., 2009; Giordan et al., 2013; Razak et al., 2013; Niculita et al., 2016, Petschko et al., 2016 ), or of ultra-resolution images acquired by Unmanned Aerial Vehicles (UAV, Niethammer et al., 2010, Giordan et al., 2015a, 2015b; Torrero et al., 2015, Turner et al., 2015).*

AC-Sincerely, we don't' understand the comment. It seems there is no request/observation.

*Line 59 > of the mentioned parameters, which are photographic characteristics, and which ones are morphological? Please explain, also considering that one of the main findings of this work is that a photographic landslide signature is best mapped with higher resolution images, while morphometric signatures are better identified with stereoscopic images.*

AC-Correct. We split the sentence in two sentences to clarify what photographical and morphological signatures are. The sentence now reads:

> "The heuristic visual mapping of landslide features is based on the systematic analysis of image photographic characteristics such as colour, tone, mottling, texture, shape, and morphological characteristics such as size and curvature, concavity and convexity (Pike, 1988). These photographic and morphological characteristics encompass all the possible landslide features that can be used for the (visual) interpretation of the available imagery."

*Line 66 > maps prepared to exploit one or more of the mentioned techniques are inevitability incomplete. Is that true? Shouldn't we affirm that they "can be" incomplete, rather than making such a strong statement?*

AC- We are aware that this statement is strong. But we underline that this is a logical consequence of the consideration that any technique or images have intrinsic limitation. If this is true, this means that these images will be somewhat "blind" for some landslides (e.g., due to the size, type, surrounding land cover), for example due to the spatial or spectral resolution, or lack of three-dimensional information. Nevertheless, we understand that such a strong clause could be better explained and supported in the text. Now, the text quoted by the Reviewer reads:

> "All these mapping techniques have inherent advantages and intrinsic limitations, which depend on the characteristics of the images, including their spatial and spectral resolutions (Fiorucci et al., 2011). The limitations affect differently the mapping, based on the size and type of the investigated landslides. As a result, images from single sources or the single mapping techniques are "blind" to some landslides, which inevitably results in incomplete landslide inventory maps. Furthermore, maps also can contain errors in terms of the position, size and shape of the mapped landslides (Guzzetti et al., 2000; Galli et al., 2008, Santangelo et al., 2015a)."

*Line 76 > 'images of different types' > 'images from different sources'*

AC- We thank R3 for this suggestion. We acknowledge the problem, and we changed the text.

*Line 77 > as I mentioned in the abstract, technically you do not compare eight maps, but rather seven maps. The dGPS is the ground truth.*

AC-We thank R3 for this suggestion. We corrected accordingly.

*Line 80 > on board OF a UAV.*

AC- We thank R3 for this suggestion. We corrected the error accordingly.

**Study area**

*Line 96 > " and A third located…"*

AC- We thank R3 for this suggestion. We corrected the error accordingly.

**Image acquisition**

*What software has been used for the SFM technique?*

AC- the software used is Agisoft Photoscan. We added this specification in the text.

*Line 124: "i.e. the same day…" does not make sense. The same day is not an example, thus, i.e. is not needed*

AC- We thank R3 for this suggestion. We removed i.e. from the sentence.

**Landslide mapping**

*Again, you do not compare eight maps. You compare seven maps and use one survey as a benchmark.*

*Line 147: i.e. is overused. 'who carried out the field activities (the reconnaissance field mapping and the RTK-DGPS survey) were not involved…"*

AC- We thank R3 for this suggestion. We removed i.e. from the sentence.

**Field mapping**

*Line 173 again overuse of 'i.e.' > the figure is not an example*

AC- We thank R3 for this suggestion. We removed i.e. from the sentence.

**Mapping through images**

*Line 201 again overuse o f i.e. > the TC and FCC images are not an example of the images used. They are indeed the images used.*

AC- We thank R3 for this suggestion. We removed i.e. from the sentence.

*Why the need of d raping the UAV image to google earth? The survey itself allows for the creation of a DSM, why using further sources (google) to interpret the images?*

AC- The reason why the ultra-high resolution UAV image was draped on Google earth is technological and is explained in the following sentence of section 4.2.:

> "To interpret visually the ultra-resolution UAV image, the interpreter overlaid ("draped") the image on Google Earth™. For the purpose, we first treated the UAV image with the gdal2tiles.py software to obtain a set of image tiles compatible with Google Earth™ terrain visualization platform. To the best of our knowledge, the platform is the only free 2.5D image visualisation environment that allows the editing of vector (point, line, polygon) information. Other commercial (e.g., ArcScene) and open source (e.g., ParaView, GRASS GIS), 2.5D visualization tools do not provide editing capabilities. Google Earth™ is a user-friendly solution for mapping single landslides, and for preparing landslide event inventories for limited areas, with the possibility for the user to visualize a landscape from virtually any viewpoint, facilitating landslide mapping. We refer to the representation of the Assignano landslide obtained through the visual interpretation of the ultra-resolution UAV image as "Map H"."

**Results.**

*Lines 23 5-238: I disagree. Visually, Fig. 5F is not that different from 5G or H: F underestimates the lower part of the landslide and misses some features in the top part. I would say that visually the most similar is 5E.*

AC- We agree that the most similar is the map E as evident also from the value of the error index (fig--). The text was changed accordingly.

**Discussion**

*Lines 318 t o 332 are not needed: they are a summary of what has already been said before. I think the whole chapter until line 397 can be shortened because much of it is a repetition o f the previous one.*

AC- We thank R3 for this suggestion. The suggestion was accepted and the text modified accordingly.

*The authors should focus more on either explaining the numbers or discussing them in general as compared to other works or examples.*

AC- We thank R3 for this suggestion. However, as stated in the introduction, the literature is rather poor in providing examples of similar studies to be compared to the present work. We cited all the papers in our knowledge:

"Our results are similar to the results of tests performed to compare field-based landslide maps against GPS-based surveys of single landslides (Santangelo et al., 2010), the visual interpretation of very-high resolution stereoscopic satellite images (Ardizzone et al., 2013), or the semi-automatic processing of monoscopic satellite images (Mondini et al., 2013), and confirm the inherent difficulty in preparing accurate landslide maps in the field, unless the mapping is supported by a GPS survey or a similar technology."

**Concluding remarks**

*Lines 457-468 are not needed. All of this has already been explained throughout the manuscript.*

AC- We thank R3 for this suggestion. We removed most of the text as suggested by the reviewer.

*Line 465-47 should also be mentioned in the study area description, explaining what photographi c and morphological signatures are.*

AC-We corrected the text according to the suggestion.

*There is no need to explain the results as a list (First….second…third), unless the authors make a short bullet point list of the main findings, and then further discuss them.*

AC-We agree, we removed the list.

*Line 512 and following. This whole part can be rephrase d without using the personal point of view (e.g. We maintain, We emphasise…' You can simply state that ' 
[revised manuscript text omitted]

[Figure]

**Figure 2**

[Figure]

**Figure 3**

[Figure]

**Figure 4**

[Figure]

**Figure 5**

[Figure]

**Figure 6**

[Figure]

**Figure 7**

[Figure]